# High-resolution and high-accuracy topographic and transcriptional maps of the nucleosome barrier

**Zhijie Chen[1,2†], Ronen Gabizon[1†], Aidan I Brown[3], Antony Lee[4], Aixin Song[5], César Díaz-Celis[1,2], Craig D Kaplan[6], Elena F Koslover[3], Tingting Yao[5]\*, Carlos Bustamante[1,2,4,7,8,9,10,11]\***

[1]Institute for Quantitative Biosciences-QB3, University of California, Berkeley, Berkeley, United States; [2]Howard Hughes Medical Institute, University of California, Berkeley, Berkeley, United States; [3]Department of Physics, University of California, San Diego, San Diego, United States; [4]Department of Physics, University of California, Berkeley, Berkeley, United States; [5]Department of Biochemistry and Molecular Biology, Colorado State University, Fort Collins, United States; [6]Department of Biological Sciences, University of Pittsburgh, Pittsburgh, United States; [7]Jason L Choy Laboratory of Single-Molecule Biophysics, University of California, Berkeley, Berkeley, United States; [8]Biophysics Graduate Group, University of California, Berkeley, Berkeley, United States; [9]Department of Molecular & Cell Biology, University of California, Berkeley, Berkeley, United States; [10]Department of Chemistry, University of California, Berkeley, Berkeley, United States; [11]Kavli Energy Nanoscience Institute, University of California, Berkeley, Berkeley, United States

**\*For correspondence:**
Tingting.Yao@Colostate.edu (TY);
carlosb@berkeley.edu (CB)

[†]These authors contributed equally to this work

**Competing interests:** The authors declare that no competing interests exist.

**Abstract** Nucleosomes represent mechanical and energetic barriers that RNA Polymerase II (Pol II) must overcome during transcription. A high-resolution description of the barrier topography, its modulation by epigenetic modifications, and their effects on Pol II nucleosome crossing dynamics, is still missing. Here, we obtain topographic and transcriptional (Pol II residence time) maps of canonical, H2A.Z, and monoubiquitinated H2B (uH2B) nucleosomes at near base-pair resolution and accuracy. Pol II crossing dynamics are complex, displaying pauses at specific loci, backtracking, and nucleosome hopping between wrapped states. While H2A.Z widens the barrier, uH2B heightens it, and both modifications greatly lengthen Pol II crossing time. Using the dwell times of Pol II at each nucleosomal position we extract the energetics of the barrier. The orthogonal barrier modifications of H2A.Z and uH2B, and their effects on Pol II dynamics rationalize their observed enrichment in +1 nucleosomes and suggest a mechanism for selective control of gene expression.
DOI: https://doi.org/10.7554/eLife.48281.001

## Introduction

The organization of genomic DNA into nucleosomes represents the main physical barrier to transcription by Pol II and constitutes a fundamental mechanism for regulation of gene expression in eukaryotes. In canonical (hereafter referred to as WT) nucleosomes, a core histone octamer made up of two copies of histones H2A, H2B, H3 and H4, is wrapped by ~147 basepairs (bp) of DNA. Variations in DNA sequence, wrapping strength asymmetry, and position-dependent histone-DNA interactions are collectively responsible for the uneven character and polarity of the nucleosomal barrier to an elongating polymerase (*Bondarenko et al., 2006*; *Hall et al., 2009*; *Ngo et al., 2015*). The

topography of the nucleosomal barrier can be described using two parameters: its height at each position (i.e., the magnitude of the energy required to access the DNA) and its width (i.e., extension along the DNA). Although Pol II has been shown to be capable of transcribing through the nucleosome both in vitro (*Bondarenko et al., 2006*) and in vivo (*Weber et al., 2014*), the detailed, high-resolution dynamics of Pol II crossing the nucleosomal barrier have not been observed yet. Because the properties of the barrier likely determine the dynamics of a transcribing polymerase, obtaining high-resolution topographic and transcriptional maps of the barrier lies at the heart of understanding the regulation of gene expression.

The majority of eukaryotic genes have a well-defined +1 nucleosome (the first nucleosome encountered by Pol II following initiation), which is enriched in H2A.Z and uH2B histones (*Rhee et al., 2014*; *Teves et al., 2014*), and represents the highest barrier to transcription (*Teves et al., 2014*). Whether the high prevalence of H2A.Z and uH2B modifies the intrinsic barrier at the +1 nucleosome, results in a different local spatial organization of chromatin, plays a role in regulating the binding and/or activity of extrinsic transcription factors, or a combination of all of these, remains unknown. Early optical tweezers studies have shown that in front of a WT nucleosome, Pol II slows down, pauses, backtracks, and cannot actively 'peel' the DNA wrapped around the histones (*Hodges et al., 2009*). Instead, the polymerase functions as a fluctuating ratchet that advances by rectifying transient, spontaneous wrapping/unwrapping transitions of the nucleosomal DNA around the histone core (*Hodges et al., 2009*). A similar study using tailless histones and mutated DNA sequences suggests that these nucleosomal elements modulate the topography of the barrier by affecting the density and duration of Pol II pauses (*Bintu et al., 2012*). However, because of their low resolution, these studies failed to accurately map the topography of the barrier and its effects on the dynamics of transcription. A high-resolution transcriptional map around the nucleosome is necessary to ultimately understand how the interaction of trans-acting factors at specific and selective positions of the polymerase around the octamer regulate transcription across the barrier.

Prior attempts to characterize the nucleosomal barrier to transcription have suffered from two substantial limitations. First, previous assessment of nucleosome stability relied on pulling and unwrapping the nucleosome from both ends (*Mihardja et al., 2006*). These experiments, while providing a measure of the strength of DNA/histone interactions, may not fully recapitulate the physical process of nucleosome invasion by Pol II, which unidirectionally unwinds the nucleosomal DNA. Second, although we can now obtain transcription trajectories with millisecond temporal and near bp spatial resolution using optical tweezers (*Righini et al., 2018*), it is very difficult to determine the absolute location of the polymerase on the template (*Hodges et al., 2009*; *Bintu et al., 2012*; *Fitz et al., 2016*), what we term here 'accuracy'. Here we surmount both limitations, obtaining high-resolution, high-accuracy topographic and transcriptional maps of WT and modified nucleosomes. By registering the dynamics of Pol II as a function of its position along the nucleosome, these maps provide a means to interrogate how variant and epigenetically modified histones affect the dynamics of transcription through the nucleosome.

## Results

### Single-molecule unzipping of nucleosomal DNA maps the topography of the nucleosome barrier

To experimentally recapitulate the underlying physical process of barrier crossing, that is nucleosomal DNA unwinding, we mimicked the effect of Pol II passage through the nucleosome using mechanical force. To this end, we adapted a previously described single-molecule DNA unzipping assay (*Hall et al., 2009*; *Rudnizky et al., 2016*) in which the two strands of the nucleosomal DNA are held in two optical traps, resulting in a Y-shaped configuration (*Figure 1A*). We engineered the stem ahead of the fork to consist of two consecutive segments of '601' nucleosome positioning sequence (NPS) (*Lowary and Widom, 1998*), and a short hairpin loop at the end to prevent tether breaking once all double-stranded DNA (dsDNA) is converted into single-stranded DNA (ssDNA) (*Figure 1A*). During each experiment, we move the two traps apart at a constant speed of 20 nm/s. When the force reaches ~17 pN, the dsDNA at the stem begins to unzip. When the stem segment does not contain a nucleosome, the DNA unzips following a series of closely spaced transitions

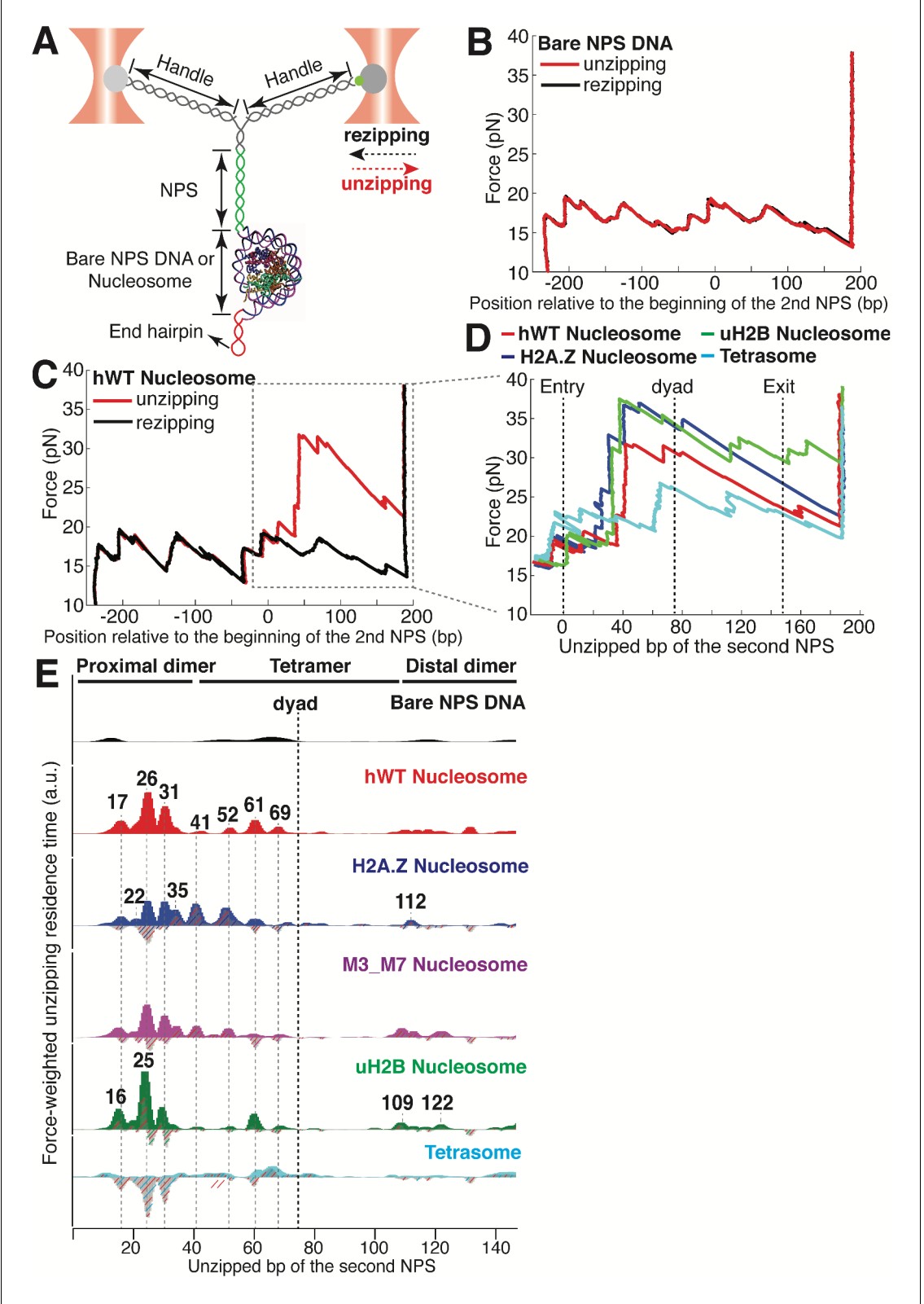

**Figure 1.** Dual-trap Optical Tweezers Single-molecule Unzipping Assay Unwinds Nucleosomal DNA and Maps Histone-DNA Interactions. (**A**) Geometry of the single-molecule unzipping assay. Dashed arrows denote directions of trap movement (20 nm/s) during unzipping (red arrow) or rezipping (black arrow). Two DNA handles connect to the template DNA, which consists of two tandem NPS repeats and an end hairpin. Diagram illustrates nucleosome unzipping, with the second NPS replaced with a pre-assembled mononucleosome. For simplicity, linkers and restriction sites flanking the

*Figure 1 continued on next page*

*Figure 1 continued*

NPS are not shown. (**B, C**) Unzipping (red) and rezipping (black) traces of bare NPS DNA (**B**) and a single WT human nucleosome (**C**). The presence of the nucleosome on the second NPS causes characteristic high force (20–40 pN) transitions that correspond to disruption of histone-DNA contacts. The unzipped basepairs (bp) are normalized to the beginning of the second NPS. The nucleosome rezipping trace matches that of bare NPS DNA, indicating complete histone removal during unzipping. (**D**) Representative unzipping traces of tetrasome (cyan), WT (red), H2A.Z (blue), and uH2B (green) nucleosomes. For clarity, only the region after entering the second NPS (corresponding to the boxed region in (**C**)) is shown, with the unzipped bp normalized to the beginning of the second NPS. The three dashed lines are entry, dyad, and exit of the second NPS, respectively. Rezipping traces, identical to those of B and C, are not shown. (**E**) Topography maps are plotted as force-weighted residence time (RT) histograms of the unzipping fork along bare NPS DNA, tetrasome and different types of nucleosomes during unzipping at constant trap separation speed of 20 nm/s. The gray histograms with colored stripes (excluding Bare NPS DNA and WT Nucleosome) are residual plots found by subtracting the WT nucleosome RTs. Unzipped bp are normalized to the beginning of the second NPS core. Major peaks are highlighted with gray dashed lines, with the peak positions (in bp) labeled above the peaks. (Left to right: 17, 22, 26, 31, 35, 41, 52, 61, 69, 109, 112, 122 bp). n = 34, 41, 34, 39, 35, 10, respectively for NPS DNA, hWT, H2A.Z, M3_M7, uH2B nucleosome and tetrasome. See also *Figure 1—figure supplement 1* for representative unzipping traces and analysis.

DOI: https://doi.org/10.7554/eLife.48281.002

The following figure supplement is available for figure 1:

**Figure supplement 1.** Unzipping Traces of Single Human WT, H2A.Z, M3_M7, uH2B Nucleosomes and Tetrasomes.

DOI: https://doi.org/10.7554/eLife.48281.003

occurring in a narrow range of forces between 17–20 pN, dictated by the sequence of the template. Once all dsDNA has been fully converted into ssDNA, the force increases sharply at the hairpin end (*Figure 1B*). The highly reproducible force-extension signatures from the two consecutive NPS regions allow us to align traces from different unzipping experiments by placing the force and dwell-time of the opening junction at each base pair into register (*Figure 1B* and *Figure 1—figure supplement 1*). Upon force relaxation (rezipping), the pattern of closely spaced transitions is reproduced in the inverse sense to that observed in the pulling direction (*Figure 1B*, *Video 1*).

Next, we repeated the experiment with the second NPS preassembled with a human WT nucleosome. The unzipping force-extension signature of the first NPS matches those obtained above, but that of the nucleosome region deviates significantly due to histone-DNA contacts (*Figure 1C* and *Figure 1—figure supplement 1A*, *Video 2*). Relaxation of the tether results in two identical, sequential rezipping signatures characteristic of the naked DNA in ~75% of the cases (*Figure 1B and C*), indicating that full DNA unzipping led to complete histone removal. However, in 25% of the cases, when we unzip the same molecule for the second time, the force reaches higher values than with bare DNA but lower than those observed the first time, likely reflecting residual histone-DNA interactions from nucleosomal relics. Since we have no knowledge of nucleosome integrity by the second round of unzipping, we only analyzed the first round of unzipping data for each molecule. Because we moved the trap at a constant speed, the dwell-time of the fork at each position reflects the local histone-DNA interaction strength at that force. Indeed, in these constant pulling velocity experiments, the forces applied to histone-DNA contacts lying deeper in the structure depend on the

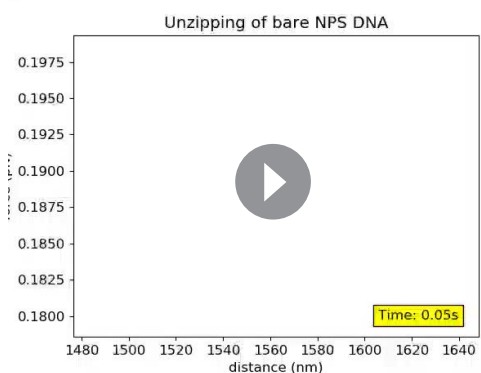

**Video 1.** Unzipping-rezipping of bare NPS DNA.

DOI: https://doi.org/10.7554/eLife.48281.004

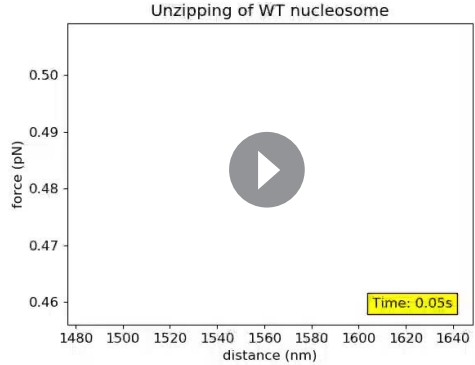

**Video 2.** Unzipping-rezipping of hWT nucleosome.

DOI: https://doi.org/10.7554/eLife.48281.007

forces reached previously in undoing earlier contacts. This effect may lead to underestimation of the magnitude of later interactions. Accordingly, we also performed force-jump unzipping experiments on the same constructs in which we suddenly increased the force to 28 pN (this force was chosen to minimize the contribution of the dsDNA sequence to the dynamics of the fork) and held it constant while monitoring the unzipping fork dwell-time at each position along the NPS (*Figure 2A*). In these experiments, the bare DNA construct unzips to the hairpin end instantaneously, while the fork dwells in the WT nucleosome primarily at 25, 31, and 35 bp into the nucleosome (*Figure 2B*). The force-weighted residence time histograms (see Methods for details) of the unzipping fork along the entire NPS obtained from these two types of experiments are similar and provide a quantitative description of the barrier to nucleosomal DNA unzipping with single bp resolution that we term the *nucleosome topography map* (*Figures 1E* and *2B*).

The topography maps reveal that the unzipping fork encounters substantial resistance at around 17, 26, 31, 41, 52, 61 and 69 bp into the nucleosome, which correspond to regions of proximal dimer and tetramer interaction with the first half of nucleosomal DNA (*Figure 1E*). Interestingly, the resistance diminishes significantly after the dyad, suggesting that unzipping the first half of the nucleosome destabilizes the histone-DNA interactions of the second half. As previously observed, major histone-DNA interactions first occur ~55 bp from the dyad and exhibit 5 or 10 bp periodicity as the unzipping fork progresses (*Hall et al., 2009*; *Rudnizky et al., 2016*). This observation probably reflects the strong histone-DNA contacts along the DNA minor groove every 10 bp observed in the crystal structure of the nucleosome (*Luger et al., 1997*). Compared to a previous study (*Hall et al., 2009*), we noticed a shorter residence time near the nucleosome dyad, which we attribute to differences in pulling geometry, buffer conditions, and/or histone source.

Unzipping of tetrasomes (H3/H4 tetramer assembled on NPS) revealed a substantially diminished barrier compared to the octamer, with unzipping fork dwelling events mostly restricted to locations near the dyad, and much lower maximum force reached during the unzipping process (*Figure 1E*, *Figure 1—figure supplement 1B*). These data indicate that the H2A/H2B dimer not only interacts locally with the DNA but it also affects the strength of the H3/H4 tetramer-DNA interaction near the dyad to orchestrate the overall nucleosome stability. As loss or exchange of H2A/H2B dimers has been implicated in important biological processes such as DNA replication (*Ramachandran and Henikoff, 2015*), transcription (*Ramachandran et al., 2017*; *Kireeva et al., 2002*), repair (*Ransom et al., 2010*), and DNA supercoiling (*Sheinin et al., 2013*); these findings highlight the potential role of non-local histone-DNA interactions in those processes.

## H2A.Z and uH2B alter orthogonal parameters of the nucleosome topography map

Human H2A.Z and uH2B nucleosomes show altered topography maps when compared to their WT counterparts (*Figures 1E* and *2B*). However, the relative magnitude and distribution of the peaks are differently affected by these two modifications. Specifically, uH2B nucleosomes stabilize the dimer region (16 and 25 bp peaks) with minor effects on the tetramer region (*Figures 1E* and *2B*), suggesting that the attachment of ubiquitin to H2B enhances the barrier height locally. The peaks after the dyad are less pronounced and correspond to regions where nucleosomal DNA interacts with the distal dimer. H2A.Z nucleosomes show enhanced peaks at 41 and 52 bp, while exhibiting much lower heights at 25 and 31 bp (*Figures 1E* and *2B*), indicating a global redistribution of the barrier's strength along its width. Overall, in H2A.Z nucleosomes the dwell-time peaks are more broadly distributed throughout the first half of the barrier than in their WT or uH2B counterparts, while maintaining their 5 to 10 bp periodicity.

To determine whether this redistribution of the barrier strength reflects features of the individual H2A.Z nucleosomes or the superposition of barriers derived from a heterogeneous molecular population resulting from an enhanced lateral mobility of these nucleosomes, as has been previously suggested (*Rudnizky et al., 2016*), we counted the number of transitions (rips) per unzipping trace in the nucleosome region. Indeed, the H2A.Z-containing nucleosome has on average one more transition per trace than its WT counterpart (*Figure 1—figure supplement 1F*) but displays a similar standard deviation. This effect is also evident in constant force unzipping experiments on H2A.Z nucleosomes, in which more dwell-time peaks are observed along the NPS (*Figure 2B*). To check whether H2A.Z nucleosomes are more mobile compared to WT nucleosomes, we repeatedly unzipped-rezipped single WT or H2A.Z nucleosomes up to the proximal dimer region (maximum

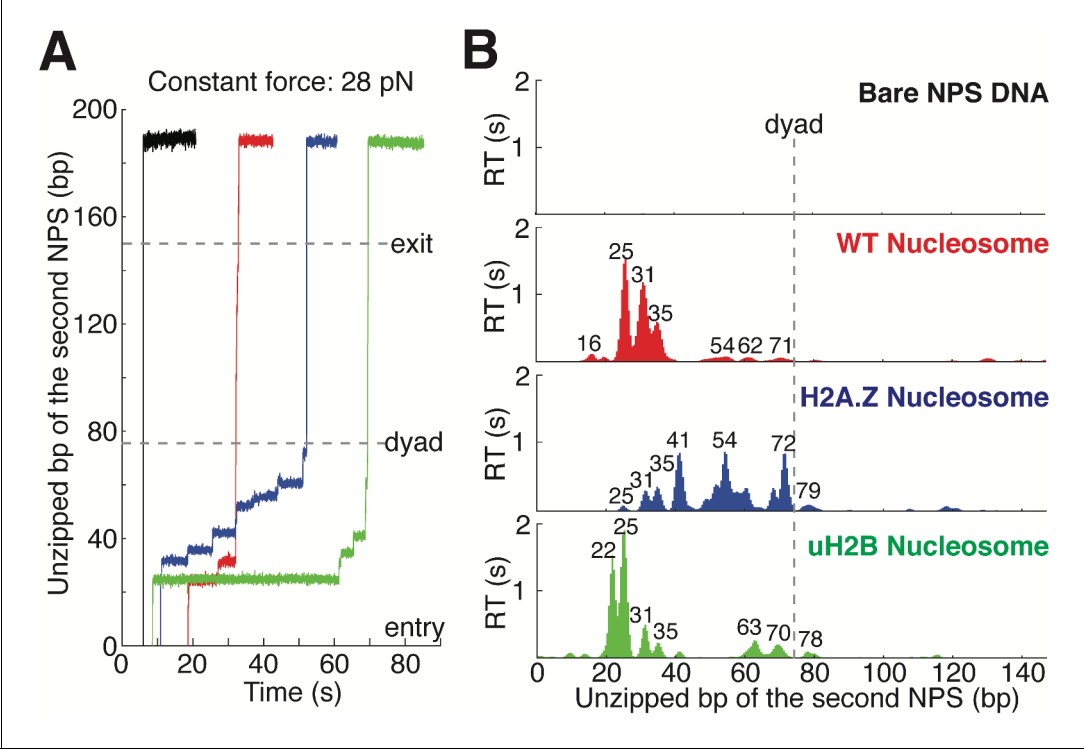

**Figure 2.** Topography Maps of the Nucleosome Revealed by Nucleosome Unzipping at Constant Force. (**A**) Representative unzipping traces of bare NPS DNA (black), WT (red), H2A.Z (blue) and uH2B (green) nucleosomes at 28 pN constant force. Unzipped bp are normalized to the beginning of the second NPS. Dashed lines mark entry, dyad and exit regions of the second NPS. Traces are shifted horizontally for clarity. (**B**) Mean residence time (RT) histogram of the unzipping fork along bare NPS DNA (black), WT (red), H2A.Z (blue) and uH2B (green) nucleosomes during unzipping at a constant force of 28 pN. Bare NPS RTs are too short to see on the axes shown. Unzipped bp are normalized to the beginning of the second NPS core. Major peak positions are indicated above each peak (in bp). n = 33, 17, 20, 20, respectively for NPS DNA, WT, H2A.Z and uH2B nucleosomes. See also *Figure 2—figure supplement 1* on assembly cooperativity of H2A.Z nucleosomes.

DOI: https://doi.org/10.7554/eLife.48281.005

The following figure supplement is available for figure 2:

**Figure supplement 1.** H2A.Z Nucleosomes Assemble More Cooperatively than WT nucleosomes.

DOI: https://doi.org/10.7554/eLife.48281.006

force reached to ~25–30 pN). If H2A.Z induces lateral mobility of the nucleosome, the position of initial force rise in the nucleosome region would shift between each partial unzipping-rezipping round, and should be quite evident in our finely registered traces. Surprisingly, in contrast to the report by *Rudnizky et al. (2016)*, we observed no lateral mobility with either WT or H2A.Z nucleosomes, as indicated by the highly reversible and overlapping unzipping signatures in the proximal dimer region (*Figure 1—figure supplement 1G and H*). Together, these results indicate that the effect of the H2A.Z histone variant in our experiments is not to increase the heterogeneity of the nucleosome population but to significantly redistribute the strength of the barrier, effectively broadening it. This conclusion is also supported by the homogeneous migration of H2A.Z nucleosomes in native gels. (*Figure 1—figure supplement 1I*).

The distal dimer interaction peaks for H2A.Z nucleosomes are visibly diminished relative to those of WT and uH2B nucleosomes (*Figure 1E*). Interestingly, we observed an increased cooperativity during the assembly of H2A.Z nucleosomes. As the ratio of octamer to DNA is increased during nucleosome reconstitution, we consistently observed significantly less hexasome formation with H2A.Z than with H2A (*Figure 2—figure supplement 1A*). It is possible that the global decrease of DNA interaction with the distal dimer observed with H2A.Z nucleosomes could also reflect a more cooperative disassembly during unzipping. To pinpoint what regions within H2A.Z are responsible for its assembly cooperativity, we generated a series of sequence swap mutants between H2A and H2A.Z (*Figure 2—figure supplement 1B*) (*Clarkson et al., 1999*). Swapping the sequences of the

M3 or the M7 region in H2A.Z with the corresponding sequences in H2A promotes the appearance of hexasomes, indicating decreased cooperativity in assembly (*Figure 2—figure supplement 1A*). Consistently, the topography map of M3_M7 nucleosomes (an M3 and M7 combined swap mutant) showed intermediate topographical features between H2A.Z and WT nucleosome, with the distal dimer interaction peaks (at 109 and 122 bp) becoming more pronounced than those of H2A.Z nucleosomes (*Figure 1E*), consistent with the idea that cooperativity in disassembly correlates with this distal interaction. Structurally the M3 region corresponds to the 'loop 1' that mediates H2A.Z-H2A.Z interactions within the octamer, and the M7 region corresponds to the 'docking domain' that mediates H2A.Z interactions with H3-H4 (*Zlatanova and Thakar, 2008*). These regions play important roles in the stability of the histone octamer. Thus, unique physical properties of the H2A.Z octamer likely account for the broadened barrier distribution we observed during unzipping.

## Observation of multiple nucleosomal states at the proximal dimer region

One unique feature from the nucleosome unzipping traces is the presence of fast, reversible unzipping transitions within the proximal dimer region—spanning the first 40 bp of the NPS—that manifest as 'hopping' of force and extension in the unzipping experiments (*Figure 3—figure supplement 1A*). Hopping in this region is nucleosome-specific, as it is not observed during unzipping of bare NPS DNA (*Figure 3—figure supplement 1B*). The reversible nature of this transition contrasts with the irreversible transitions observed deeper in the nucleosome, and may indicate a rapid, small scale unzipping that is not large enough to disrupt the structure of the octamer. With nucleosome unzipping occurring at approximately 20–30 pN (*Figure 1C–D*), to observe hopping at our trap separation rate of 20 nm/s (with trap spring constant of approximately 0.3 pN/nm) requires the system to transition between states on timescales of approximately 2 s or less. To better capture these hopping dynamics, we fixed the trap distance such that the unzipping fork remained localized within the proximal dimer region and monitored the fluctuations of the force and extension (passive mode experiment). Within an empirically determined trap distance range, we obtained equilibrium extension hopping traces (*Figure 3A*), which show transitions on timescales of ~1 s, aligning with the necessary rate of transitions between states. At each fixed trap separation, we determined the number of unzipped bp to obtain a probability distribution for the length of unzipped DNA (*Figure 3B*). For both bare and nucleosomal DNA, these distributions show consistent peaks, as expected on a heterogeneous energy landscape where the system populates discrete energy wells separated by transition barriers. We note that certain trap separations allow the system to sample multiple wells, giving rise to a multi-modal distribution (*Figure 3B*, bottom panel), analogous to the hopping observed in the constant pulling rate unzipping curves. Relative to bare DNA, WT nucleosomes display an additional peak in the distribution of unzipped bps, at approximately 28 bp after the start of the second NPS where the most significant contacts between DNA and the H2A-H2B dimer occur (*Figure 3C*). This peak implies the existence of a barrier to further unzipping that arises from binding of the DNA to histones, and its position is consistent with the dwell time peak observed in the unzipping traces (*Figure 1E*, peak at 26 bp).

Assuming that the observed distributions (*Figure 3B and C*) correspond to equilibrium Boltzmann statistics, we extracted the energy associated with unzipping of each bp in the proximal dimer region of the nucleosome (*Figure 3D*, details in Methods). The presence of a strong interaction energy peak at 32 bp into the WT nucleosome and the corresponding energy well preceding this new barrier position account for the appearance of the new preferred state in the distribution of unzipped bps (*Figure 3C*, peak at 28 bp). Furthermore, the low barrier to rezipping from this state (* in *Figure 3D*) implies that the dynamics between the two states in the proximal dimer region of the nucleosome (at approximately 18 bp and 28 bp) should be quite rapid, in keeping with the hopping behavior observed in the unzipping curves (*Figure 3—figure supplement 1A*). Subtracting the unzipping energy of bare DNA from that of the nucleosome, we obtained the additional energy associated with each nucleosome type, providing a measure of the interaction energy between the DNA and the octamer throughout the first quarter of the NPS (*Figure 3D*, inset). WT, H2A.Z, and uH2B nucleosomes all show strong DNA binding to the proximal dimer region, with a large peak in the interaction energy centered at approximately 35 bp.

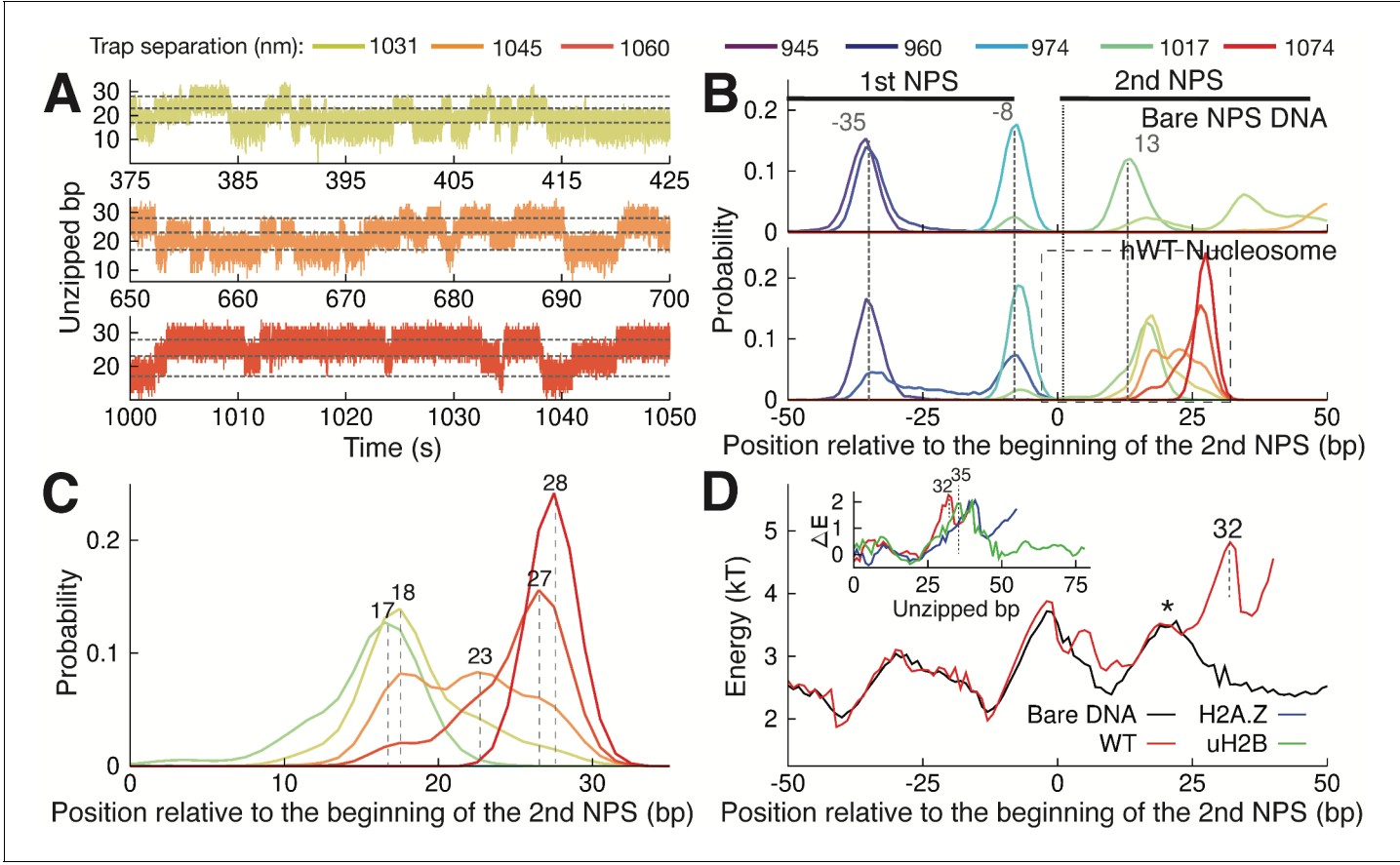

**Figure 3.** Observation of Multiple Nucleosomal States at the Proximal Dimer Region. (**A**) Time traces of number of base pairs unzipped (relative to beginning of the second NPS) with hWT nucleosome for fixed trap separations of 1031 nm, 1045 nm, and 1060 nm (top to bottom). Color indicates increasing trap separation (purple to red), corresponding to clusters in *Figure 3—figure supplement 1F*. Gray dashed lines indicate 17, 23, and 28 base pairs unzipped. (**B**) Probability distributions for the number of DNA bps unzipped, computed from force-extension data in *Figure 3—figure supplement 1F*. Each curve is from a different trap separation, matching colors in A and *Figure 3—figure supplement 1F*. Distributions are shown for both bare DNA (top) and WT nucleosome (bottom). Vertical black dotted line indicates the start of the second NPS. Vertical gray dashed lines indicate peak positions for bare DNA (with position in bp labeled), showing that WT nucleosome shifts the first peak within the NPS, and gives rise to an additional peak at 28 bp. See *Figure 3—figure supplement 1F* for force-extension data. (**C**) Zoomed-in view of the black dashed box in (**B**). Peak positions are labeled in bp. (**D**) DNA unzipping energy computed by assuming the unzipped bp distributions from data in *Figure 3—figure supplement 1F* (including distributions in B) correspond to equilibrium Boltzmann statistics. Inset ΔE shows the DNA-octamer interaction energy, computed as the difference between unzipping energies in the presence of WT (red), H2A.Z (blue), and uH2B (green) nucleosomes and unzipping energies for bare DNA only (black). Vertical black dashed lines and * indicate peak positions (labeled in bp). See also *Figure 3—figure supplement 1* on hopping traces and analysis of energy landscape from equilibrium data.

DOI: https://doi.org/10.7554/eLife.48281.008

The following figure supplement is available for figure 3:

**Figure supplement 1.** Hopping of the Unzipping Fork Near the Proximal Dimer Region of the Nucleosome.
DOI: https://doi.org/10.7554/eLife.48281.009

## A 'molecular ruler' gauges the positions of an elongating pol II with near Base-pair accuracy

Having established the topography of the nucleosomal barrier via mechanical force-induced DNA unwinding, we next set out to determine how this topography manifests in the dynamics of Pol II during transcription through the nucleosome. We used a high-resolution dual-trap optical tweezers instrument together with an improved nucleosomal transcription assay (*Figure 4*). To accurately gauge the positions of Pol II on the template, we placed a 'molecular ruler' in front of the nucleosome (*Figure 4A*). The 'molecular ruler' consists of eight tandem repeats of an artificially designed 64 bp DNA that has a single well-defined, sequence-encoded pause site when transcribed by Pol II

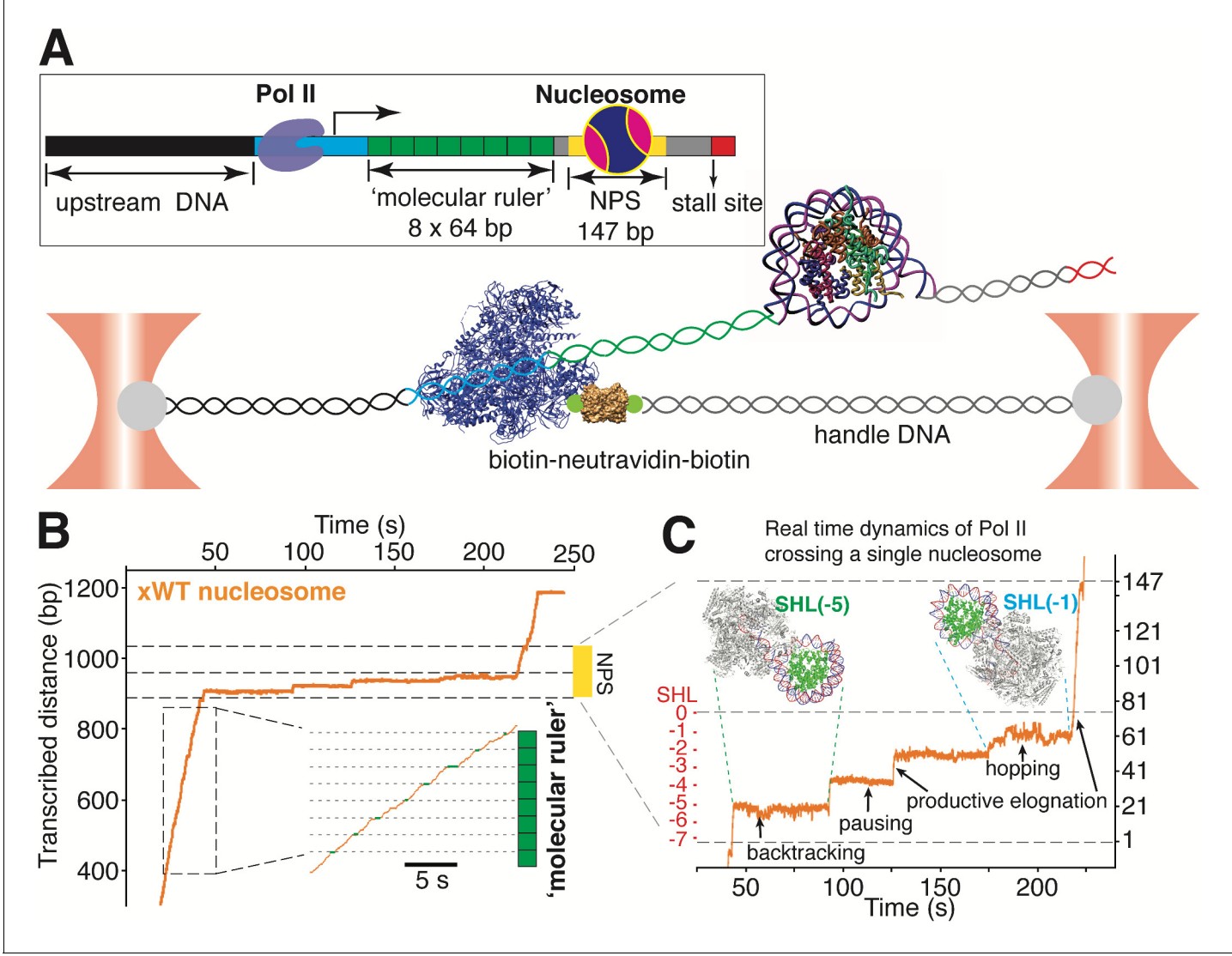

**Figure 4.** A 'Molecular Ruler' Gauges the Positions of an Elongating Pol II with Near-Basepair Accuracy. (**A**) Experimental design of an improved single-molecule nucleosomal transcription assay. A single biotinylated Pol II (purple molecular structure) is tethered between two optical traps. Pol II transcription is measured as increases in distance between the two beads at 10 pN constant force. The inset box shows the composition of the template, which is constructed by ligating Pol II stalled complex (cyan), the molecular ruler (green), NPS DNA (or nucleosome, yellow-gray), and a short inter-strand crosslinked DNA (for stalling Pol II, red). The 'molecular ruler' consists of eight tandem repeats of a 64 bp DNA (green), each harboring a single sequence-encoded pause site. (**B**) A representative trace of a single Pol II transcribing through a *Xenopus* WT nucleosome. The three black dashed lines indicate NPS entry, dyad and NPS exit, respectively. The inset shows a zoomed-in view of the boxed region, highlighting the repeating pause patterns within the 'molecular ruler'. The gray dashed lines are the predicted pause sites, whereas the short green lines mark the actual pauses of Pol II. (**C**) Zoomed-in view of Pol II dynamics within the NPS region of (**B**). The three black dashed lines indicate NPS entry, dyad and NPS exit, respectively. The right y-axis (in bp) is normalized to the beginning of the NPS. The left y-axis shows regions preceding the dyad as SHL in red. Black arrows indicates representative events of backtracking, pausing, productive elongation, and hopping. Regions corresponding to Pol II located at SHL (−5) and SHL(−1) are indicated with green and cyan dashed lines, with the corresponding Pol II-nucleosome complex structures plotted on top (PDB 6A5P for PolII-SHL(−5), 6A5T for PolII-SHL(−1)). Pol II, histones, template DNA, non-template DNA are colored in gray, green, red and blue, respectively. See also *Figure 4—figure supplement 1* on detailed characterization of the 'molecular ruler'.
DOI: https://doi.org/10.7554/eLife.48281.010

The following figure supplement is available for figure 4:

**Figure supplement 1.** Biochemical and Single-molecule Characterization of the 'Molecular Ruler'.
DOI: https://doi.org/10.7554/eLife.48281.011

in bulk (*Figure 4—figure supplement 1A*, dashed rectangle) and in single-molecule assays (*Figure 4B* and *Figure 4—figure supplement 1B–D*). The repeating pausing patterns of Pol II within the 'molecular ruler' generated a periodicity of 21.1 ± 0.3 nm (*Figure 4—figure supplement 1B*), corresponding to the length of 64 bp DNA under experimental force and buffer conditions. This periodicity serves to align all transcription traces (*Gabizon et al., 2018*; *Herbert et al., 2006*) and it also enables the accurate conversion of nanometer distances to basepairs of transcribed DNA (*Figure 4—figure supplement 1D*).

We used a bubble initiation method to assemble a stalled biotinylated yeast Pol II elongation complex (*Hodges et al., 2009*; *Bintu et al., 2012*) that was ligated downstream to a 2 kb spacer DNA and upstream to the 'molecular ruler', followed by a single nucleosome (*Figure 4A*). The '601' NPS was used to ensure both precise nucleosome positioning and accurate assignment of Pol II positions as it crosses the barrier. Pol II transcription was restarted by supplying a saturating concentration of NTPs (0.5 mM). A Pol II stall site consisting of a short inter-strand cross-linked DNA segment was placed downstream of the NPS (*Figure 4A*). In these assays, we used force-feedback to maintain a constant 10 pN assisting force throughout the transcription trajectory so that the increase of the distance between the beads serves as an accurate measure of how far Pol II has transcribed. We find this tethering geometry to be superior to prior designs (*Hodges et al., 2009*; *Bintu et al., 2012*) because it isolates Pol II from the beads surfaces, thus mitigating photo-damage (*Landry et al., 2009*). A representative real-time trajectory of Pol II transcribing through the 'molecular ruler' followed by bare NPS DNA is shown in *Videos 3* and *4*.

## Real-time, High-resolution Dynamics of Single Pol II Enzymes Transcribing Through Single Nucleosomes

We first obtained traces of Pol II transcribing through bare NPS DNA (*Figure 5A and B*, black trace). Utilizing the pausing patterns obtained with the 'molecular ruler' (*Figure 4B*, inset), we adapted a recently described algorithm (*Gabizon et al., 2018*) to align the traces such that the positions of Pol II along the entire NPS are under registry (*Figure 4B* and *Figure 4—figure supplement 1D*). The results show that Pol II has a median crossing time within the NPS of 11 s (*Figure 6—figure supplement 1A*), a pause free velocity of 28.9 ± 3.0 nt-s$^{-1}$ (*Figure 6—figure supplement 1H*), and displays very few backtracking events (0.17 per trace) (*Supplementary file 2*).

Next, we replaced the bare NPS DNA with an assembled *Xenopus* WT (xWT) nucleosome on the same template (*Videos 5* and *6*). As expected, Pol II exhibits a dramatic slow-down, with a median crossing time of 129 s (*Figure 6—figure supplement 1B*), a pause-free velocity of 11.4 ± 4.1 nt-s$^{-1}$ (*Figure 6—figure supplement 1H*), and frequent pausing and backtracking within the NPS region (*Figure 4C*). 59% of Pol II succeeded in crossing the barrier (*Figure 6—figure supplement 1F*), which is signaled by its reaching the stall site at the end of the template (*Figure 4B*). Using the 'molecular ruler' to precisely locate Pol II on the template, and after subtracting the enzyme's footprint (16 bp) (*Shao and Zeitlinger, 2017*), we obtained a median dwell-time histogram of the leading edge of Pol II along the entire NPS with ±3 bp accuracy (depicted as linear and polar plots in *Figure 6A*). This dwell-time histogram, which we refer to as a *transcriptional map* of the nucleosome, illustrates at high resolution and accuracy the height and width of the nucleosome barrier to the elongating Pol II. It complements the topographic map described above, translating it into a 'functional map'. Our measurements represent a nearly 20-fold resolution and accuracy improvement on previous attempts to obtain a transcriptional map of the nucleosomal barrier, since those experiments could only resolve roughly three barrier regions of ~50 bp each, corresponding to entry, central and exit

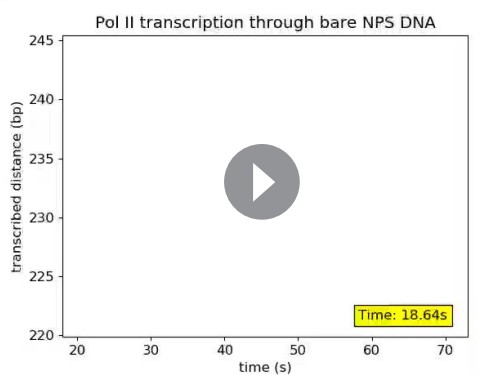

**Video 3.** Pol II transcription through bare NPS DNA. The horizontal gray dashed lines indicate predicted pause sites in the molecular ruler, the three horizontal black dashed lines represent NPS entry, dyad, and NPS exit, respectively. This applies to all other videos.
DOI: https://doi.org/10.7554/eLife.48281.012

zones (*Hodges et al., 2009*; *Bintu et al., 2012*; *Fitz et al., 2016*).

Several features of Pol II barrier crossing dynamics emerge from the nucleosome transcriptional map (*Figure 6A*). First, the effect of the barrier begins immediately after the leading edge of Pol II touches the NPS (3 bp peak); the strength of the barrier is largest at 28 bp and 10–20 bp before the dyad, and is negligible after crossing this pseudo-symmetry axis. Second, we identified a region between 28–64 bp into the NPS where Pol II enters long-lived pauses and backtracks frequently (*Figure 4C*). These pauses are consistent among different molecules and exhibit a ~ 10 bp periodicity (28, 38, 48, 57, and 64 bp into the NPS). Notably, this region coincides with the region of maximum resistance in the single-molecule unzipping assay (*Figures 1E* and *2B*), implying that the transcriptional barrier encountered by Pol II while crossing the nucleosome reflects, to a first approximation, the barrier mapped by the unzipping assay. Third, some molecules were permanently arrested in this region (*Figure 5A,B*, gray trace), but those that managed to cross it typically succeeded in reaching the stall site shortly thereafter (*Figure 5A*). Thus, we speculate that this region (28–64 bp) may play an important regulatory role for barrier crossing by the enzyme. The observed asymmetry of the transcriptional map between both sides of the dyad axis (*Figure 6A*) may reflect a substantial weakening of the histone-DNA interactions in the presence of the bulky resident enzyme halfway across the barrier. However, the transcriptional map asymmetry is similarly observed in the topographic map (*Figure 1E*), even though the bulkiness of the enzyme does not play a role in those experiments. It is also possible that the barrier asymmetry reflects changes in the nucleosome integrity by the invading polymerase (*Kireeva et al., 2002*) or the propagating unzipping fork.

*Xenopus* histones are traditionally the most-widely used in nucleosome studies because they are well behaved in recombinant form. Since we employed recombinant human histones in the unzipping assays, it was of interest to compare the Pol II transcriptional maps of *Xenopus* nucleosomes with those obtained with their human counterparts utilizing the same 601 NPS. As seen in *Figure 6A*, the maps are quite similar except that human nucleosomes confer a significantly higher barrier to transcription (see *Figure 5A–B*, *Videos 7* and *8* for representative traces) than those of *Xenopus* in the proximal dimer region (*Figure 6A*, the 28 bp peak in orange and red panels).

## Dynamic interplay between pol II and the nucleosome during barrier crossing

Interestingly, we observed extensive two state transition dynamics while Pol II is paused at certain sites (frequently at 28 and 63 bp) (*Figure 4C*, *Video 6*). While Pol II hopping at 28 bp coincides with hopping of the unzipping fork near this region and probably reflects sampling of alternative nucleosomal states ahead of the enzyme, hopping at a much deeper location into the nucleosome (63 bp) may have a more complex origin. We hypothesize that these latter dynamics may be due to local Pol II-histone interactions or re-wrapping of the nucleosome in front of a backtracked enzyme. Indeed, these hopping dynamics occur exclusively after Pol II enters a deeply backtracked state (*Figure 5— figure supplement 1*).

We also investigated whether Pol II remains functionally competent after the crossing. The pause-free velocity of Pol II after exiting the nucleosome resumed to 70% ($28.6 \pm 0.8$ nt-s$^{-1}$) of its value before the crossing ($41.1 \pm 1.0$ nt-s$^{-1}$). This observation seems to indicate that while the enzyme remains functionally competent, its dynamic state has been affected by the encounter (*Figure 6— figure supplement 1H*). It remains unknown what changes in the enzyme are responsible for this slowing down and if they are reversible. On the other hand, we probed the integrity of the nucleosome by pulling away the two beads after Pol II crossed the barrier. If the nucleosome survived the traversal by Pol II, it would now lie between the two tethering points, that is the upstream DNA handle and the polymerase. Rarely (<5%) these pulling curves displayed the force-extension signature normally associated with the presence of a nucleosome, suggesting that in situ reassembly of the nucleosome following Pol II traversal was inefficient under our assay conditions. A similar low efficiency of nucleosome reassembly was observed from transcription assays in bulk in the absence of factors added in trans such as FACT (facilitates chromatin transcription) (*Hsieh et al., 2013*).

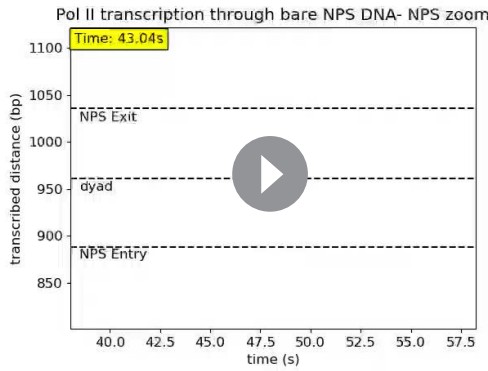

**Video 4.** Pol II transcription through bare NPS DNA, NPS zoom.
DOI: https://doi.org/10.7554/eLife.48281.013

## H2A.Z enhances the width and uH2B the height of the transcriptional barrier

Next, we investigated the effects of human H2A.Z and uH2B on Pol II transcription dynamics. Our unzipping assay revealed that H2A.Z and uH2B have distinct effects on the nucleosome barriers (*Figures 1E* and *2B*). Under the same buffer and force conditions, Pol II alone was capable of crossing either H2A.Z (*Figure 5C and D*, *Videos 9* and *10*) or uH2B (*Figure 5E and F*, *Videos 11* and *12*) nucleosomes. While the crossing probabilities of Pol II through H2A.Z and WT nucleosomes are similar (58% and 59%, respectively) (*Figure 6—figure supplement 1F*), the distributions of pause sites within the NPS are markedly different (*Figure 6A*) in that H2A.Z is seen to cause a global redistribution of Pol II pause sites along the entire NPS. Such scattered distribution (3, 28, 36, 59, 66, 87, 101, 115, 125 and 138 bp) differs significantly from that of WT nucleosome (3, 28, 57, 64 bp) where most pauses occur before the dyad. This spreading of the barriers for H2A.Z nucleosomes is a further indication that the force applied to the upstream DNA (which is the same in both experiments) is not the dominant factor responsible for the asymmetry of the WT barrier across the dyad, but that the actual histone-DNA interactions are. Furthermore, these differences are unlikely to stem from H2A.Z nucleosomes being mis-positioned on the starting template, because the first pause site at ~3 bp, where the leading edge of Pol II begins to interact with the nucleosome, is observed in every molecular trajectory obtained with H2A.Z nucleosomes. Instead, we attribute them to broadened distributions of DNA-histone interactions as seen in the topography map for H2A.Z nucleosomes (*Figures 1E* and *2B*, also see discussion on energetic profiles of H2A.Z nucleosome in next section). Clearly, H2A.Z strongly modulates the width of the nucleosomal barrier to transcription.

In contrast, Pol II transcription through human uH2B nucleosomes has a slightly higher passage probability (76%) than through hWT nucleosomes (*Figure 6—figure supplement 1F*), although at the expense of longer crossing times (*Figure 6—figure supplement 1E*). The pause site distribution also resembles that of the hWT or the xWT nucleosomes (*Figure 6A*), however, the median dwell-time of Pol II at pause sites near the dyad is more than double that for the WT nucleosome, suggesting that uH2B enhanced the height of the nucleosomal barrier to Pol II (*Figure 6A*). The overall pause-free velocities of Pol II transcription through H2A.Z ($11.4 \pm 7.1$ nt-s$^{-1}$) and uH2B nucleosomes ($12.9 \pm 3.4$ nt-s$^{-1}$) are lower than that through hWT ($18.2 \pm 7.1$ nt-s$^{-1}$) nucleosomes (*Figure 6—figure supplement 1H*). Consequently, the median crossing times of Pol II through H2A.Z (262 s) and uH2B (304 s) nucleosomes are longer than that through hWT nucleosomes (230 s) (*Figure 6—figure supplement 1C–E*). Note, however, that pause-free velocity contributes negligibly to crossing time, as the latter is dominated by long pauses. Including pausing, translocation and backtracking, Pol II takes longer to cross uH2B or H2A.Z than hWT nucleosomes (*Figure 6B*).

Similar to when traversing hWT nucleosomes, Pol II backtracks frequently during transcription through H2A.Z and uH2B nucleosomes (*Figure 5—figure supplement 1A,C and E*). The average number of backtracks and backtrack depths are similar, but backtrack durations are longer when Pol II transcribes through H2A.Z nucleosomes than through WT and uH2B counterparts (*Supplementary file 2*), which may again reflect the broader extent of histone-DNA interactions in H2A.Z nucleosomes. Failure to recover from some backtracks seems to contribute to Pol II arrests, the positions of which were more scattered for Pol II transcribing through uH2B and H2A.Z than through WT nucleosomes (*Figure 6—figure supplement 1G*). Interestingly, backtracked Pol II frequently enters long-lived pauses, some of which are also accompanied by frequent two-state transition dynamics (hopping behavior) (*Figure 5—figure supplement 1D and F*). During some of the long-lived pauses associated with crossing of uH2B nucleosomes, we also observed three-state

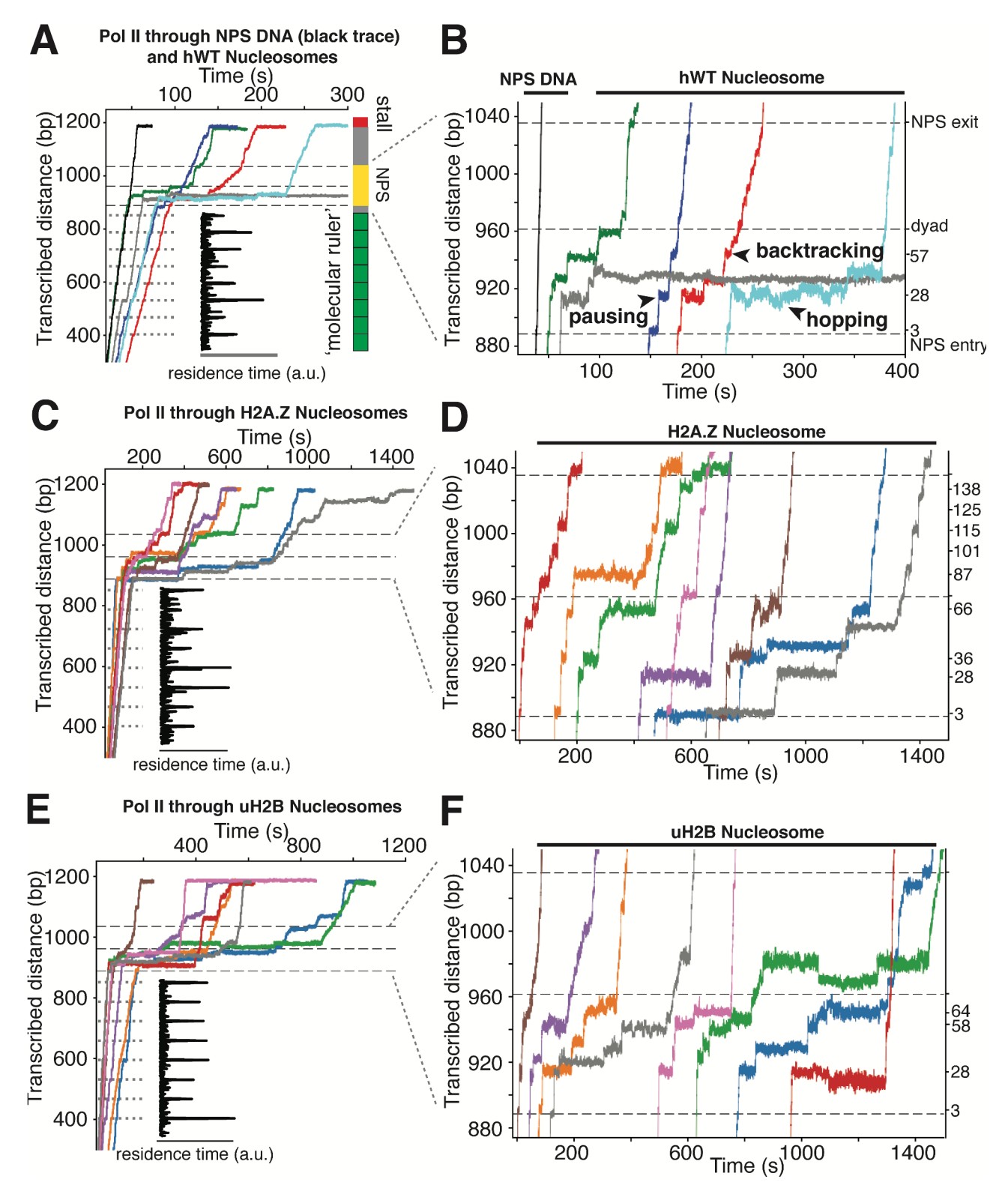

**Figure 5.** High-resolution Trajectories of Individual Pol II Enzymes Transcribing through WT, H2A.Z and uH2B Nucleosomes. (**A**, **B**) Representative traces of single Pol II enzymes transcribing through single human WT nucleosomes. The gray dotted lines are the pause sites within the 'molecular ruler'. The inset (black) is the residence time of Pol II within the 'molecular ruler', highlighting repeating pausing signatures of Pol II. The three black dashed lines indicate NPS entry, dyad and NPS exit. Relative positions of Pol II on the template DNA are shown as a cartoon on the right. The traces in *Figure 5 continued on next page*

*Figure 5 continued*
blue, green, red and cyan are examples of successful nucleosome crossing, while the trace in gray is an example of Pol II arrest in the nucleosome. For comparison, a trace of Pol II transcribing through bare NPS DNA (black) is shown on the left. Zoomed in traces of high-resolution Pol II dynamics within the NPS are shown in (B), highlighting (black arrowheads) long-lived pausing, backtracking and hopping events. The traces are shifted horizontally for clarity. The right y-axis is normalized to the beginning of the NPS, with the major pause positions marked (in bp) on the right. (C, D) Representative traces of single Pol II enzymes transcribing through single human H2A.Z nucleosomes. (C) shows the full traces and (D) is a zoomed-in view of the high-resolution dynamics within the NPS region. (E, F) Representative traces of single Pol II enzymes transcribing through single human uH2B nucleosomes. (E) shows the full traces and (F) is a zoomed-in view of the high-resolution dynamics within the NPS region. See also *Figure 5—figure supplement 1* on backtracking and hopping dynamics.
DOI: https://doi.org/10.7554/eLife.48281.014
The following figure supplement is available for figure 5:

**Figure supplement 1.** Long-lived Pauses of Pol II in the Nucleosome are Associated with Backtracking and Hopping Dynamics.
DOI: https://doi.org/10.7554/eLife.48281.015

hopping behavior (*Figure 5—figure supplement 1F*) and large hopping transition events (*Figure 5F*, green trace) of Pol II. As this behavior is rarely observed in WT and H2A.Z traces (*Figure 5*), we speculate that these dynamics of Pol II reflect the presence of the bulky ubiquitin attachment. Collectively, these data reveal that H2A.Z mainly enhances the width and uH2B mainly enhances the height of nucleosomal barrier to transcription. Consistent with the previously reported higher stability of uH2B nucleosomes (*Batta et al., 2011*; *Chandrasekharan et al., 2009*), they pose an overall higher barrier magnitude—especially in the region near the dyad—than their WT counterparts to the passage of polymerase. It is also worth noting that for WT and H2A.Z, but not for uH2B nucleosomes, the transcriptional map replicates the corresponding topography map, suggesting that there could be uH2B-Pol II specific interactions that are not present in the unzipping assay.

Our findings provide direct evidence that H2A.Z or uH2B by themselves affect the crossing dynamics of Pol II; and despite their differential effects on nucleosome topography, they both represent stronger barriers than WT nucleosomes for Pol II.

## A mechanical model for pol II transcription through the nucleosome

We use a simplified mechanical model to calculate the expected polymerase dwell times along nucleosomal DNA given a profile of DNA-octamer interaction energies. We build on a previously published Pol II model that includes a mechanical DNA linkage between Pol II and the nucleosome, in addition to nucleosome-Pol II steric interactions (*Figure 7A*) (*Koslover and Spakowitz, 2012*). The model assumes that Pol II is unable to actively separate the DNA from the surface of the octamer. Instead, the enzyme behaves as a ratchet that makes progress by rectifying the unwrapping fluctuations of the nucleosomal DNA. The enzyme can also backtrack and diffuse forward to re-engage the 3'-end of the nascent transcript with its active site. The extended model incorporates varying binding energies for the DNA along the nucleosome.

Polymerase progress along the nucleosomal DNA is modeled as a series of transcription steps and backtracking excursions (*Figure 7B*), adapting the model of *Dangkulwanich et al. (2013)* to include interactions with the nucleosome. The individual polymerase steps are assumed to occur on an energy landscape that encompasses both the elastic energy of deforming the unwrapped DNA linker between Pol II and the nucleosome core particle, and the interaction energy between the wrapped DNA and the core particle (*Figure 7C*). For a given Pol II position, the amount of unwrapped DNA is assumed to fluctuate rapidly around an energy minimum that balances these two contributions. As Pol II steps forward, the linker shortens and

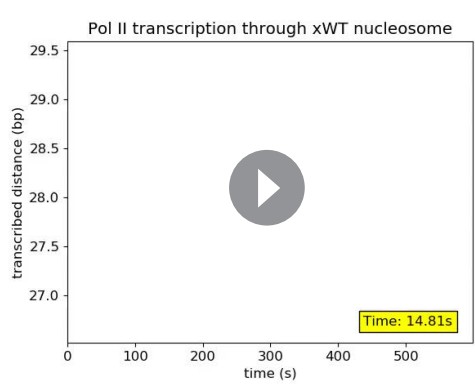

**Video 5.** Pol II transcription through *x*WT nucleosome.
DOI: https://doi.org/10.7554/eLife.48281.018

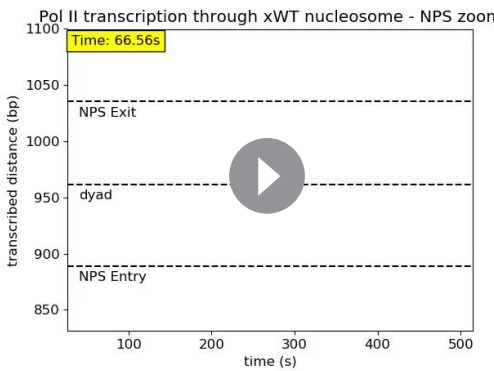

**Video 6.** Pol II transcription through xWT nucleosome, NPS zoom.

DOI: https://doi.org/10.7554/eLife.48281.019

the elastic energy increases, with a longer linker entailing a smaller energy increase and thus a more rapid rate constant for polymerase forward motion. The interaction energy profile between the DNA and the histone octamer ahead of Pol II determines the ensemble of linker lengths and, thus, indirectly controls the average time spent by Pol II at each DNA base pair. This model allows us to calculate the expected dwell time of the polymerase at each position.

Accordingly, we use the DNA-octamer interaction energies extracted from equilibrium unzipping data (*Figure 3D*) to calculate the expected mean dwell times of polymerase in the proximal nucleosomal region. The dwell time peaks resulting from these equilibrium interaction energies approximately correspond to peaks in the experimental dwell time profiles (*Figure 7—figure supplement 1*). The DNA-octamer interaction energies cannot account for the first dwell time peak (the peak at ~3 bp in *Figure 6A*) corresponding to the initial encounter between Pol II and the nucleosome. We hypothesize that this peak is the result of additional interactions between Pol II and the histone proteins rather than arising from the difficulty of peeling DNA from the nucleosome.

Because equilibrium interaction energy data was available only for the initial section of the nucleosome (*Figure 3*), Pol II dwell times further into the nucleosomal sequence cannot be predicted from the data available. Instead, we solve the inverse problem: given the measured Pol II mean dwell times (*Figure 7D*), we fit the DNA-nucleosome interaction energies (*Figure 7E*, details in Methods) required to generate this dwell time profile. The dependence of dwell times on the nucleosome binding energy in this model is non-local—time spent at a particular bp depends on the energy required to unwrap a segment of DNA *ahead* of the polymerase. Consequently, peaks in the dwell time profile arise from interactions that involve both large energy values and span a substantial length of DNA (extended regions of strong binding). For example, the peaks at 29 bp in the dwell time profiles for hWT, uH2B, and H2A.Z nucleosomes (*Figure 7D*), correspond to large peaks in the interaction energy at 32 bp, consistent with the interaction energy profiles obtained from equilibrium DNA unzipping data (*Figure 3D*). A two-peaked region of strong binding at roughly 43 bp in the H2A.Z and uH2B nucleosomes gives rise to corresponding double peaks in the dwell time profiles 41 bp into the nucleosomes, with substantially longer pausing times for H2A.Z nucleosomes in this region. An additional broad region of strong binding is seen just before the dyad axis, 62 bp in hWT and uH2B nucleosomes, resulting in the observed Pol II pausing peaks 59 bp into the nucleosomes. Interestingly, according to our mechanical model, the predicted interaction energies necessary to generate the observed dwell-time profiles are similar in magnitude for all three nucleosomal types. However, wider peaks of strong binding give rise to increased pause durations in the uH2B transcriptional profile, while a distribution of many narrower peaks accounts for the increased number of pausing sites in the H2A.Z profile (*Figure 7D,E*).

## Discussion

For the last 20 years the crystal structure of the nucleosome (*Luger et al., 1997*) has guided our view of the packaging unit of the genome and suggested its role as a regulator of gene expression. As a mechanical and energetic barrier, the nucleosome gates the accessibility of genomic DNA, constituting a fundamental regulatory mechanism for all DNA-templated processes including replication, transcription, repair, recombination, and chromatin remodeling. Epigenetic modifications and histone variants are known to modulate all of these processes. The question of whether this modulation results from the recruitment of trans-acting factors, or responds to changes in the intrinsic properties of the barrier, or both, has not previously been addressed.

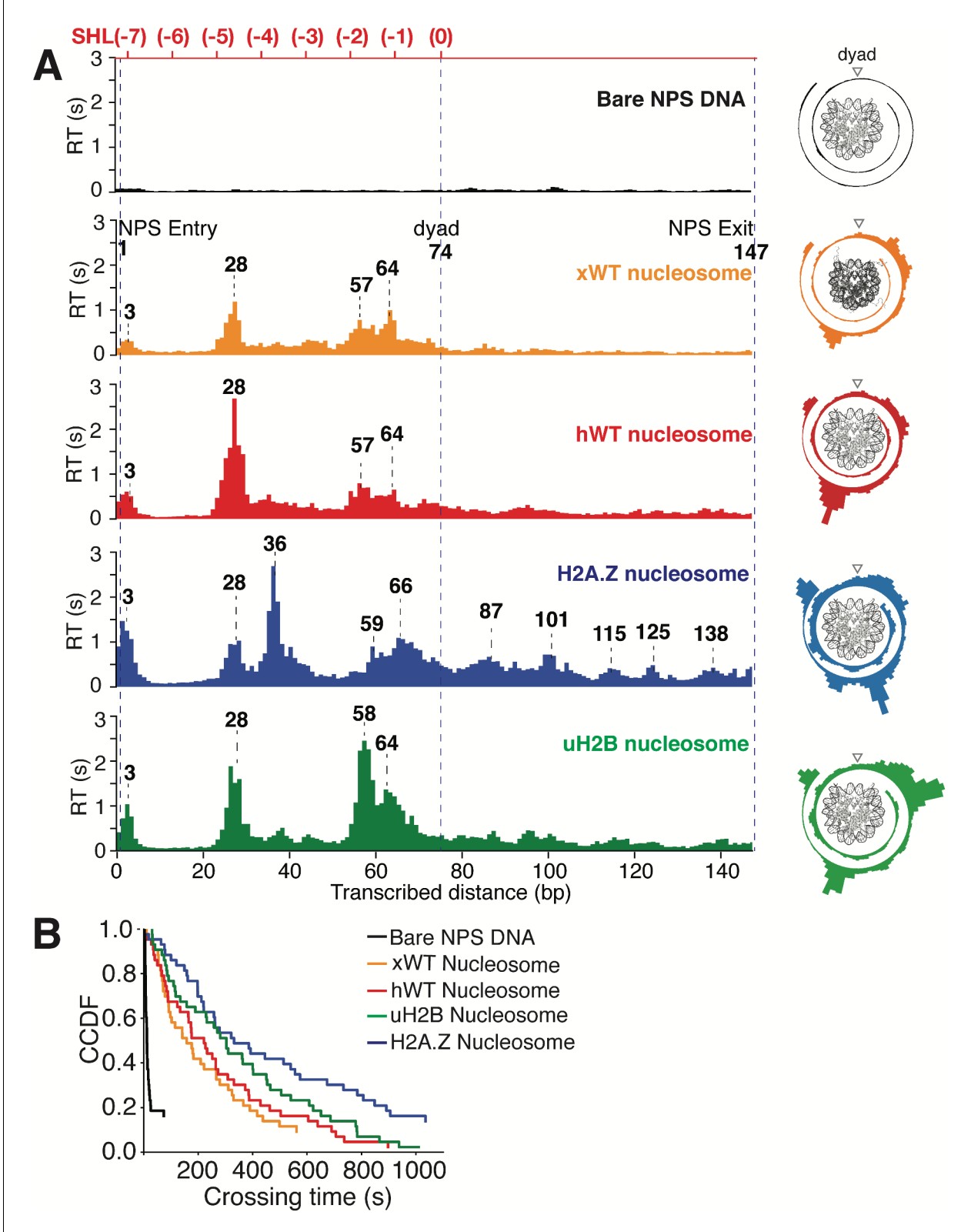

**Figure 6.** Transcriptional Maps of the Nucleosome Reveal that H2A.Z Enhances the Width and uH2B the Height of the Barrier. (**A**) Median residence time histograms of Pol II transcription through bare NPS DNA (black), xWT (orange), hWT (red), H2A.Z (blue) and uH2B (green) nucleosomes. Bar width is 1 bp and major peak positions are labeled (in bp) above the corresponding peaks. NPS entry, dyad, NPS exit are marked with blue dashed lines. The polar plots on the right are the corresponding transcriptional maps of the nucleosome, formed by projecting the residence time histogram onto the

*Figure 6 continued on next page*

*Figure 6 continued*

surface of nucleosomal DNA. The top axis (red) indicates corresponding positions of the first half of nucleosome expressed as superhelical locations (SHL). n = 35, 23, 26, 21, 31, respectively for NPS DNA, xWT, hWT, H2A.Z and uH2B nucleosomes. (**B**) Crossing time (total time Pol II takes to cross the entire nucleosome region) distributions plotted using the complementary cumulative distribution function (CCDF, fraction of events longer than a given crossing time). Crossing times of Bare NPS DNA, *Xenopus* WT (xWT), human WT (hWT), uH2B and H2A.Z nucleosomes are plotted in black, orange, red, green and blue, respectively. See also *Figure 6—figure supplement 1* on statistics of the crossing time, crossing probability, pause-free velocity and arrest position.

DOI: https://doi.org/10.7554/eLife.48281.016

The following figure supplement is available for figure 6:

**Figure supplement 1.** Crossing Time, Crossing Probability and Pause-free Velocity of Pol II during Transcription through NPS DNA or Nucleosomes.

DOI: https://doi.org/10.7554/eLife.48281.017

Low-resolution single molecule assays showed that it is possible to follow molecules of Pol II as they cross the nucleosomal barrier (*Hodges et al., 2009*; *Bintu et al., 2012*). However, these studies only yielded gross features of the barrier and failed to provide the crucial spatial-dependent dynamics of the crossing that are required to rationalize the effect of nucleosome modifications at the molecular level.

Very recently, cryo-EM structures of Pol II-nucleosome complexes have provided snapshots of Pol II paused at major histone-DNA contacts and suggested sites of interaction with other factors (*Farnung et al., 2018*; *Kujirai et al., 2018*; *Ehara et al., 2019*). Missing from these structures is information about the dynamics of barrier crossing by the enzyme: what are the time windows available for in-trans interactions with these discrete sites, how are these related to the local energetic magnitude of the barrier, and how are they modulated by epigenetic modifications and histone variants. Using a 'molecular ruler', we have been able to locate individual Pol II molecules along the template with high precision and to extract their molecular trajectories as they transcribe through nucleosomes at near bp resolution and accuracy. These trajectories unveil unprecedented details on the general dynamics (translocating, pausing, hopping and backtracking) as well as the residence times of the enzyme at every position as it progresses through the nucleosome, providing insights into how gene expression is regulated spatially and temporally at a single nucleosome level.

Our results reveal that the proximal dimer region of the nucleosome (~28 bp) in the transcription direction is a major physical barrier for Pol II and may serve as an important regulatory checkpoint for gene expression. In this region, Pol II frequently enters long-lived pauses; this result is consistent with the observation of a major Pol II pause site at the superhelical location SHL(−5) reported recently (*Kujirai et al., 2018*). Interestingly, pausing at this location is accompanied by extensive hopping dynamics, likely reflecting unwrapping/rewrapping of the nucleosomal DNA around the octamer and/or structural rearrangements of the nucleosome. Indeed, partially unwrapped nucleosomal intermediates have been detected in vitro by time-resolved small angle X-ray scattering (*Chen et al., 2014*), by cryo-EM (*Bilokapic et al., 2018*) and in vivo by MNase-seq (*Ramachandran et al., 2017*). The location of these structural intermediates coincides with the different nucleosomal hopping states observed as the unzipping fork reaches the proximal dimer, reinforcing the interpretation that local DNA/histone interactions determine the dynamics of Pol II in this region and its ultimate progress beyond it.

Traditionally, the dyad has been viewed as the strongest histone-DNA contact point and therefore as the highest barrier position in WT nucleosomes. Unambiguously assigning Pol II's residence time with bp resolution has allowed us to define a transcriptional map of the barrier, which indicates that the proximal dimer region and not the dyad represents the highest barrier

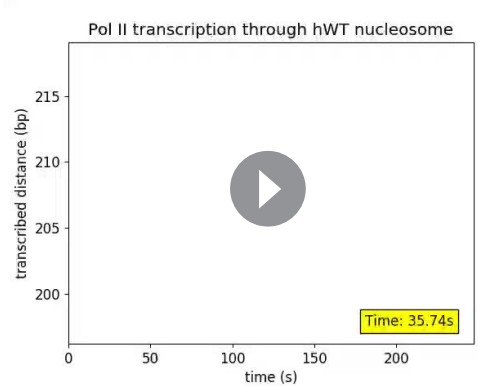

**Video 7.** Pol II transcription through hWT nucleosome.
DOI: https://doi.org/10.7554/eLife.48281.020

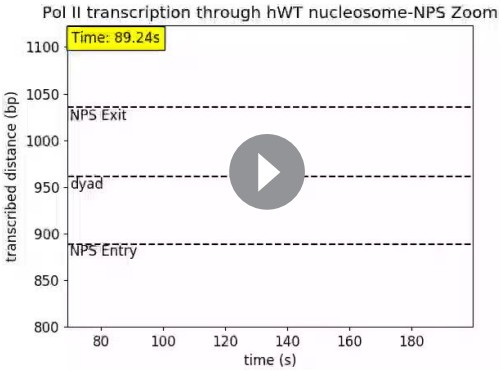

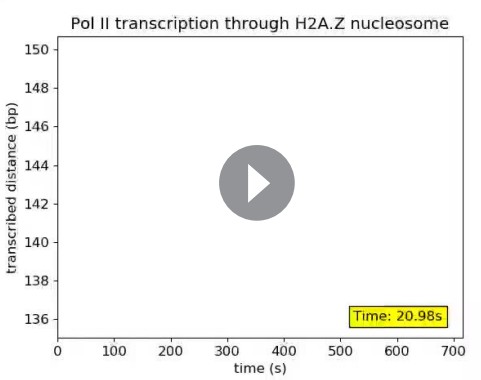

**Video 8.** Pol II transcription through hWT nucleosome, NPS zoom.
DOI: https://doi.org/10.7554/eLife.48281.021

**Video 9.** Pol II transcription through H2A.Z nucleosome.
DOI: https://doi.org/10.7554/eLife.48281.022

to an elongating Pol II. This observation is consistent with the unzipping experiments that also reveal the proximal dimer region as mechanically the most stable. We posit here that the change in dynamics of the polymerase, progressing slowly in this region, provides a crucial time window to allow for other facilitative or inhibitory factors to bind and further modulate the strength of the barrier to the transcribing enzyme. For instance, FACT may bind to the nucleosome and remove one histone dimer ahead of the Pol II and reassemble the nucleosome after Pol II traversal (*Belotserkovskaya et al., 2003*; *Mason and Struhl, 2003*; *Chen et al., 2018*). These early regulatory steps as Pol II invades the nucleosome not only gate gene expression but also permit the regulation of chromatin integrity and of epigenetic modifications. As Pol II progresses further into the nucleosome, the strength of the barrier appears to be dynamically modified either through nucleosome destabilization, the steric bulkiness of the enzyme, or both. Beyond the dyad, there is practically no barrier in WT nucleosomes, again in agreement with a recent cryo-EM structural report (*Kujirai et al., 2018*).

Consistent with these observations, modifications that play important regulatory roles such as H2A.Z and uH2B, mainly affect the proximal dimer region, although their effects are not circumscribed to this location. Pol II transcription through nucleosomes bearing H2A.Z or uH2B reveal that these modifications strongly increase the strength of the barrier, but do so distinctively: H2A.Z increases the width of the barrier whereas uH2B increases its height. Significantly, the topographic map of the WT and H2A.Z barriers before the dyad, as determined here by force-induced nucleosomal DNA unwinding, closely parallels the transcriptional map of Pol II, indicating that to a first approximation, the ability of the enzyme to cross the barriers in this region is dictated by the energetic requirements of disrupting DNA-histone interactions.

In vitro, H2A.Z has been observed to either enhance or decrease nucleosome stability depending on the assays used (*Zlatanova and Thakar, 2008*; *Bönisch and Hake, 2012*). Our improved optical tweezers experiments offer unprecedented resolution that captures a more complex picture in which H2A.Z redistributes the strength of the barrier across and beyond the dyad, effectively increasing its width. Accordingly, the physical barriers across the H2A.Z nucleosome are lower, yet more globally distributed. The precise origin of this broader distribution is not known. A previous study suggested

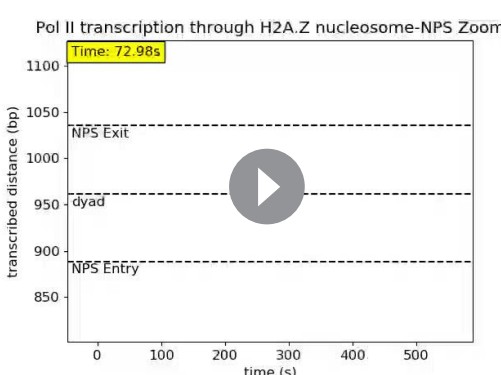

**Video 10.** Pol II transcription through H2A.Z nucleosome, NPS zoom.
DOI: https://doi.org/10.7554/eLife.48281.023

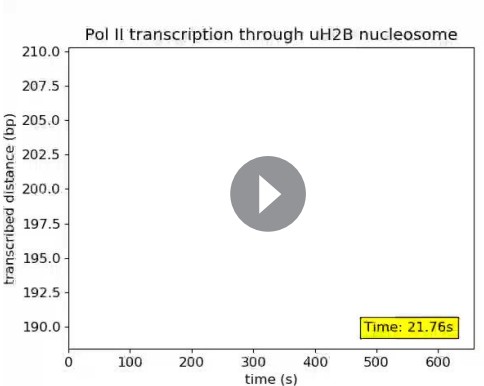

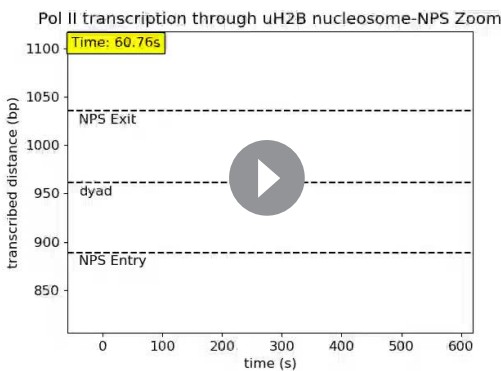

**Video 11.** Pol II transcription through uH2B nucleosome.
DOI: https://doi.org/10.7554/eLife.48281.024

**Video 12.** Pol II transcription through uH2B nucleosome, NPS zoom.
DOI: https://doi.org/10.7554/eLife.48281.025

that H2A.Z nucleosomes are more mobile compared to their WT counterparts, although the extent and cause of the mobility remain unclear (*Rudnizky et al., 2016*); however, we do not observe this enhanced mobility in our experiments. Like a previous study (*Park et al., 2004*), our data support the idea that the H2A.Z octamer is more stable than its WT counterpart within the nucleosome. As a result, H2A.Z hexasomes are barely observed during nucleosome assembly and we find that the M3 and M7 regions within H2A.Z are important for conferring such increased octamer stability. We speculate that increased octamer stability strengthens the overall nucleosomal barrier as reflected in the increased crossing time of Pol II through such nucleosomes.

The effect of H2A.Z on the nucleosomal barrier can be seen as that of re-distributing the strength of the barrier from height to width. Interestingly, the Arrhenius dependence of barrier crossing time predicts that the time to cross a barrier of height n x h is proportional to the nth-power of the time to cross a barrier of height h. In contrast the time to cross n sequential barriers of height h is proportional to n times the time to cross each one of the barriers. Thus, based on these considerations alone, we would expect that H2.A.Z would decrease the crossing time of Pol II, not increase it, as observed. Therefore, factors other than barrier crossing time, but favored by the presence of the barrier (e.g. backtracking, pausing, etc), are the ones that dominate the overall crossing time in the case of H2A.Z nucleosomes. By dividing the height of the barrier into several smaller ones, H2A.Z nucleosomes provides the enzyme more opportunities at different locations to pause, backtrack and possibly interact with regulatory factors acting in trans such as chaperones and chromatin remodelers. In vivo, the effects of H2A.Z on transcription are complex and somewhat species-dependent. The strong barrier posed by H2A.Z nucleosomes may explain its role in poising quiescent genes for activation in yeast (*Santisteban et al., 2011*) and its prevalence in +1 nucleosomes across eukaryotic genomes (*Weber et al., 2014*). In contrast, the observation that H2A.Z facilitates transcription in multi-cellular organisms (*Weber et al., 2014*) is more likely due to recruitment of trans-acting factors.

Using homogeneous, chemically-defined recombinant nucleosomes, we also demonstrated that uH2B strengthens histone-DNA interactions at the dimer region and increases the overall barrier strength to Pol II. Interestingly, while uH2B occurs at the dimer region, its effect on Pol II transcription propagates to other regions of the nucleosome including the region preceding the dyad. Thus, the effects of epigenetic modifications are not merely local but may extend further into the barrier. Uncovering such position-dependent nucleosome properties and dynamics has been possible by the high resolution and accuracy achieved in our single-molecule assays.

In vivo, H2B ubiquitination is highly dynamic and both the addition and removal of ubiquitin are required for optimal transcription (*Wyce et al., 2007*). Like H2A.Z, it is not known whether these phenotypes are due to altered nucleosome stability or to impaired or facilitated recruitment of trans-acting factors. Nevertheless, higher levels of H2A.Z and uH2B are observed in transcriptionally silent gene promoters in yeast (*Batta et al., 2011*; *Zhang et al., 2005*). Preventing H2B ubiquitination in yeast led to increased Pol II occupancy and transcription from quiescent promoters

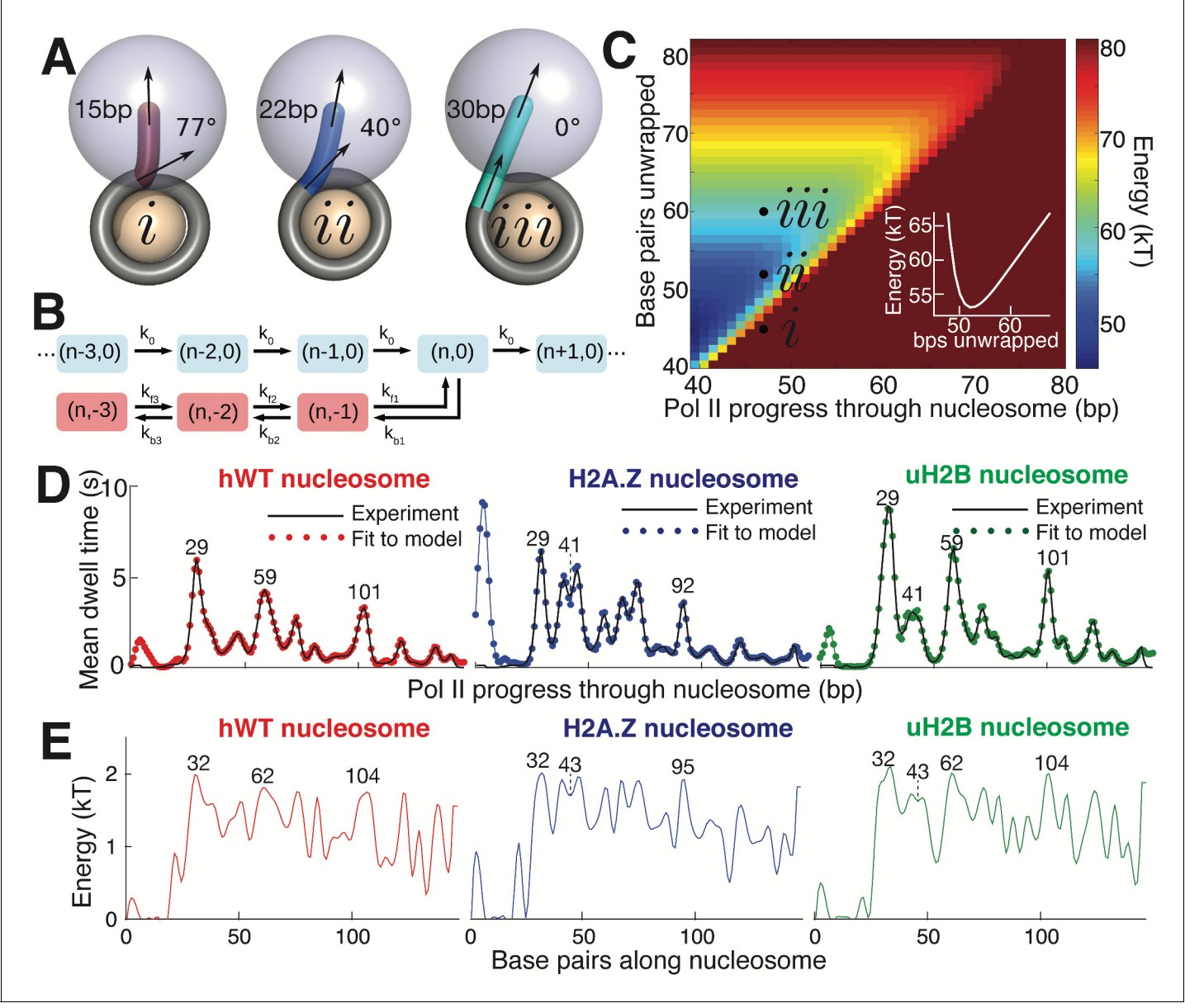

**Figure 7.** Mechanical Model for Pol II Transcription Through the Nucleosome. (**A**) Schematic of the mechanical model, showing three different lengths of unwrapped DNA for a given polymerase position along the DNA sequence. The steric spheres are shown in purple (polymerase) and beige (nucleosome), while the DNA is shown as a tube. (**i**) shows a configuration with a short, sharply bent DNA linker connecting Pol II and the nucleosome, which are in contact and sterically pushing on each other. (**ii**) shows a medium-length straighter linker, with Pol II still pushing on the nucleosome. (**iii**) shows a long straight linker without contact between Pol II and the nucleosome. Linker DNA color corresponds to overall energy for each configuration (given in C). Black arrows represent tangent orientations of the DNA backbone at the point of polymerase binding (top) and for the last contact with the nucleosome (bottom). Linker length and bending angle (between indicated tangents) are labeled on each polymerase-nucleosome pair. (**B**) Model of Pol II dynamics. Pairs (**p,q**) indicate the Pol II state: p indicates the length of the RNA transcript, and q the number of base pairs backtracked from the most recent main pathway state. Pol II steps forward one base pair with rate $k_0$ or can enter a backtracked pathway by stepping backward one base pair at rate $k_{b1}$. From backtracked positions, Pol II can move forward a base pair with rate $k_{fn}$ or can backtrack another base pair at rate $k_{bn}$. Moving forward from the first backtracked state returns Pol II to the main pathway. (**C**) Energy landscape of nucleosome-Pol II interaction, for constant DNA-nucleosome interaction energies of $1k_BT$ per base pair. DNA unwrapping decreases the DNA linker conformational energy, while removing favorable DNA-nucleosome interactions, overall providing a minimum energy a few base pairs ahead of the front edge of Pol II. Forward Pol II steps are unfavorable as they shorten the DNA linker. Points *i*, *ii*, and *iii* correspond to configurations illustrated in A. Inset shows cross-section of energy landscape at Pol II position of 47 bp, highlighting the minimum in the energy landscape a few bps ahead of Pol II, at ~52 bps unwrapped. Pol II progress through the nucleosome is defined as the position of the Pol II center plus an additional 17 bp for consistency with the transcribed distance in *Figure 6*. (**D**) Dwell time profiles for human WT, H2A.Z, and uH2B nucleosomes. Solid black lines are experimental mean dwell times and colored

*Figure 7 continued on next page*

*Figure 7 continued*

dotted lines are the best fitted mean dwell times according to the mechanical model. (E) Estimated DNA-octamer interaction energy profiles for human WT, H2A.Z, and uH2B nucleosomes. The energy values are found such that they give the best fitted dwell times shown in (D). Peak positions referenced in the text are labeled in bp, relative to the start of the NPS. See also *Figure 7—figure supplement 1* for fitting of nucleosome energy profiles based on Pol II dwell times.

DOI: https://doi.org/10.7554/eLife.48281.026

The following figure supplement is available for figure 7:

**Figure supplement 1.** Fitting Nucleosome Energy Profiles Based on Pol II Dwell Times.

DOI: https://doi.org/10.7554/eLife.48281.027

(*Batta et al., 2011*). These observations are consistent with our data that H2A.Z and uH2B provide orthogonal and selective means to enhance the transcription barrier thus contributing to the maintenance of the transcriptional ground state and to gene silencing. Our results show that while these modifications may also act indirectly through their actions on the binding of trans-acting factors, they exert a direct and significant effect on transcription dynamics by Pol II.

We have developed a unified mechanical model that uses the experimentally determined space-resolved residence times of the enzyme at each position on the nucleosome to determine the energetics of the barrier. This model is quite general and should prove useful in predicting the behavior of Pol II through alternative barriers, and in understanding the mechanics of barrier crossing for other molecular motors.

The dynamics of Pol II transcription through the nucleosome in vivo are affected by numerous other factors such as higher-order chromatin folding, DNA topology, and transcription regulators including histone chaperones, elongation factors, and chromatin remodelers. Integrating one or more of these elements in single-molecule assays such as the one presented here provides an interesting avenue for future work to fully elucidate the features and principles underlying this biologically crucial and biophysically complex molecular encounter. Because epigenetic modifications are potent regulators of eukaryotic gene expression, these results shed new light on the mechanistic link between modifications enriched on the +1 nucleosome and the barrier to transcription. More broadly, the real-time characterization of the dynamics of Pol II molecules traversing through nucleosomes at the highest resolution and accuracy reported so far, and the resulting nucleosome transcriptional map, constitute important steps towards uncovering the physical mechanisms underpinning the regulation of eukaryotic gene expression.

# Materials and methods

**Key resources table**

| Reagent type (species) or resource | Designation | Source or reference | Identifiers | Additional information |
|---|---|---|---|---|
| Antibody | NeutrAvidin (deglycosylated native avidin from egg whites) | Thermo Fisher | Cat# PI31000 | powder dissolved in PBS, 0.5 µM |
| Antibody | Anti-Digoxigenin from sheep | Sigma-Aldrich | Cat# 11333089001, RRID:AB_514496 | powder dissolved in PBS, 0.2 mg/ml |
| Strain | DH5α competent cells | Fisher Scientific | Cat# 18-265-017 | |
| Strain | BL21(DE3) competent cells | NEB | Cat# c2527H | |
| Strain | Agilent SURE 2 Supercompetent Cells | Fisher Scientific | Cat# 200152 | |
| Chemical compound | dNTP set 100 mM Solutions | Fisher Scientific | Cat# R0181 | |

*Continued on next page*

*Continued*

| Reagent type (species) or resource | Designation | Source or reference | Identifiers | Additional information |
|---|---|---|---|---|
| Chemical compound | NTP set 100 mM Solutions | Fisher Scientific | Cat# R0481 | |
| Chemical compound | 3'-Deoxyadenosine-5'-Triphosphate | TriLink Biotechnologies | Cat# N3001 | |
| Chemical compound | 3'-Deoxyguanosine-5'-Triphosphate | TriLink Biotechnologies | Cat# N3002 | |
| Chemical compound | 3'-Deoxycytidine-5'-Triphosphate | TriLink Biotechnologies | Cat# N3003 | |
| Chemical compound | 3'-Deoxyuridine-5'-Triphosphate | TriLink Biotechnologies | Cat# N3005 | |
| Chemical compound | Trioxsalen | Sigma-Aldrich | Cat# T6137 | |
| Chemical compound | $[\alpha-32P]$-ATP | Perkin Elmer | Cat# BLU003H250UC | |
| Commercial assay or kit | T4 DNA ligase | NEB | Cat# 0202L | |
| Commercial assay or kit | *E. coli* DNA ligase | NEB | Cat# 0205L | |
| Commercial assay or kit | Phusion high-fidelity DNA polymerase | NEB | Cat# M0530S | |
| Commercial assay or kit | DraIII-HF | NEB | Cat# R3510S | |
| Commercial assay or kit | BsaI-HF | NEB | Cat# R3535L | |
| Commercial assay or kit | BglI | NEB | Cat# R0143S | |
| Commercial assay or kit | EagI-HF | NEB | Cat# R3505S | |
| Commercial assay or kit | SapI | NEB | Cat# R0569S | |
| Sequence-based reagent | Lambda DNA | NEB | Cat# N3011S | |
| Recombinant DNA reagent | Primers for making constructs and DNA templates | This paper | IDT custom order | *Supplementary file 1* |
| Recombinant DNA reagent | pGEM-3z/601 | Addgene | Cat# 26656 | |
| Recombinant DNA reagent | pGM3z-8×repeat-2×BsaI | This paper | N/A | |
| Software, algorithm | LabVIEW VIs | *Comstock et al. (2011)* | RRID:SCR_014325 | |
| Software, algorithm | Matlab scripts for data analysis | This paper | RRID:SCR_001622 | |
| Software, algorithm | Python scripts for data analysis | This paper | RRID:SCR_008394 | |
| Software, algorithm | ImageJ | NIH (open source) | RRID:SCR_003070 | |
| Other | Pierce Streptavidin Magnetic Beads | Thermo Fisher | Cat# 88816 | |
| Other | BD 1 mL Insulin Syringe with Slip Tip | Fisher Scientific | Cat# 14-823-434 | |

*Continued on next page*

*Continued*

| Reagent type (species) or resource | Designation | Source or reference | Identifiers | Additional information |
|---|---|---|---|---|
| Other | Dual-trap time-shared high resolution optical tweezers | (*Comstock et al., 2011*) | N/A | |
| Other | Multi-channel optical tweezers chamber | This study | N/A | |
| Other | 1 µm carboxylated polystyrene beads | Bang Laboratories, Inc. | Cat# SVP-10–5 | |
| Other | 1.26 µm streptavidin polystyrene beads | Spherotech | Cat# PC04001-PC04N | |
| Other | Streptavidin Coated Magnetic Particles | Spherotech | Cat# SVM-08–10 | |
| Other | Hi-Trap Q HP columns 5 × 1 mL | Genesee Scientific | Cat# 17-1153-01 | |
| Other | Ni-NTA Agarose | Qiagen | Cat# 30210 | |
| Other | HisTrap HP 1 × 5 mL | Genesee Scientific | Cat# 84–208 | |
| Other | Amicon Ultra-0.5 Centrifugal Filter Unit, 100K | Millipore Sigma | Cat# UFC500324 | |
| Other | Amicon Ultra-0.5 Centrifugal Filter Unit, 3K | Millipore Sigma | Cat# UFC510024 | |
| Other | Zyppy Plasmid Maxiprep Kit | Zymo Research | Cat# D4028 | |
| Other | Typhoon imager | GE Healthcare | TRIO Variable Mode | |

## General materials

All DNA modifying enzymes were purchased from New England Biolabs (NEB). Oligonucleotides were purchased from Integrated DNA Technology (IDT). Nucleotide triphosphates were purchased from Thermo Fisher Scientific, and standard salts and buffer components were purchased from Sigma Aldrich. Cloning and DNA template construction follows standard molecular biology techniques unless otherwise noted. The sequences of all oligos used are listed in *Supplementary file 1*.

## DNA constructs for single-molecule unzipping experiments

DNA arms of the Y structure are prepared by standard PCR reactions using lambda DNA as the template. The left (with BsaI site) and right arm (with biotin) of the Y was amplified using oligos ZC01-ZC02 and ZC03-ZC04, respectively. The left arm was digested with BsaI, and annealed with the right arm to form the Y. The length of the left (after ligation) and right arm dsDNA are 937 and 911 bp, respectively.

The first (for alignment) and second (for loading) NPS were amplified with ZC05-ZC06 and ZC07-ZC08 respectively from the pGM3z-601 plasmid. The first NPS fragment was digested with BsaI, and the second NPS fragment was digested with BsaI/DraIII. The first NPS was ligated with the Y to form Y-alignment. The second NPS was ligated with the end hairpin to form NPS-hp. The end hairpin was pre-folded by heating oligo ZC09°C to 98°C for 2 min, followed by a slow decrease of temperature to 25°C at 1 °C/min. Y-alignment were purified using agarose gel. NPS-hp was purified using native PAGE followed by electro-elution and anion exchange chromatography with HiTrap-Q column.

## Beads preparation

To couple oligonucleotides to polystyrene beads, ZC10 was hybridized to ZC11 to generate a double stranded oligo containing a phosphorylated 5'-CGGT overhang. Annealing was performed by heating a 1:1 mixture of the oligos in water (0.25 mM each) to 95°C for 10 min, followed by cooling to room temperature. This resulted in the following oligo duplex:

```
5' NH2-TTAATTCATTGCGTTCTGTACACG 3'
3' TTAAGTAACGCAAGACATGTGCTGGC-phos 5'
```

1 μm diameter carboxylated polystyrene beads were coupled to the prepared double-stranded duplex as follows: 10 μL of 10% (W/V) beads were washed four times with 200 μL coupling buffer (0.1 M MES, pH = 4.7, 150 mM NaCl, 5% DMSO), and dispersed in 20 μL coupling buffer. All centrifugations took place for 5 min at 4500 g. 10 μL of 20 μM double stranded oligo and 6 μL of 2 M 1-ethyl-3-(3-dimethylaminopropyl) carbodiimide (EDC) were added, followed by vigorous shaking for 2 hr at room temperature. At this point another 10 μL of 2M EDC were added, followed by overnight shaking at room temperature. The remaining active EDC was then quenched by adding 2.5 μL of 1 M glycine, and the beads were washed 5 times with storage buffer (Tris 20 mM pH = 8, 1 mM EDTA, 0.05% Tween 20, 5 mM NaN$_3$) with 3 min of sonication between washes. The beads (1 μm oligo beads) were finally dispersed at a concentration of 1% (w/v) and stored at 4°C.

The beads were passivated by diluting 6-fold in TE (20 mM Tris, pH 8.0, 1 mM EDTA) and addition of β-casein to a final concentration of 1 mg/ml. The beads were vortexed for 10 min, washed once with TE, dispersed to a concentration of 0.2% (w/v) in TE and stored at 4°C until the experiment.

## Histone octamer assembly and purification

Recombinant human histones H2A, H2B, H3.3 and H4 were purchased from the Protein Expression and Purification (PEP) Facility at Colorado State University. H2A.Z, H2A/H2A.Z swap mutants and all *Xenopus laevis* histones were expressed in *E. coli*, purified and reconstituted into octamers according to standard protocols (*Dyer, 2003*). uH2B was prepared by crosslinking ubiquitin (G76C) and H2B (K120C) as previously described (*Long et al., 2014*).

## Nucleosome reconstitution on NPS-hp

Purified human histone octamers or tetramers were stored in 10 mM Tris, pH 7.6, 1.6 M NaCl, 1 mM EDTA, 1 mM DTT, 20% glycerol at −80°C. The reconstitution of nucleosome was performed using a salt dilution method as described (*Dyer, 2003*). Briefly, NPS-hp DNA and histone octamers or tetramers were mixed in different molar ratios ranging from 1:0.8 to 1:1.4 and initial salt concentration of the mixture (10 μL) was brought to 2 M NaCl. These reactions were incubated at 30°C in a PCR machine and the following amounts of dilution buffer (10 mM Tris, pH 7.6, 1 mM EDTA, 1 mM DTT, 0.1 mg/mL BSA) were added every 15 min: 3.3, 6.7 5, 3.6, 4,7, 6.7, 10, 30, 20, 100 μL. The reaction products were analyzed by native polyacrylamide gel electrophoresis (4%, acrylamide:bisacrylamide ratio 37.5:1) with 0.5 × TBE plus 5 mM MgCl$_2$ on ice. The reaction that gave no aggregates and minimal amounts of free DNA was chosen for further concentration using Amicon Ultra centrifugation filters with Ultracel 100K membrane. Concentrated nucleosomes were supplemented with 0.02 % NP40 and stored at 4°C.

## Optical tweezers assay for single-molecule unzipping

Unzipping oligo beads were prepared by ligating 5'-CGGT 1 μm polystyrene oligo beads with Y-alignment DNA, NPS-hp nucleosome (or NPS-hp DNA) using *E. coli* DNA ligase. The reaction was carried out at 16°C for 2–3 hr. The ligated beads were diluted with TB50 buffer (20 mM Tris, pH 8.0, 50 mM KCl, 5 mM MgCl$_2$, 1 mM DTT, 10 mM NaN$_3$, 0.1 mg/mL BSA) to a final bead density of 0.00006% (w/v) and loaded into the tweezing chamber, which was filled with TB50 buffer. Tweezing chamber was pretreated with 5% Pluronic F-127 and 1 mg/mL BSA followed by washing with TB50 buffer prior to each experiment. The 1.26 μm SA beads were diluted directed with TB50 buffer to the same bead density as that of oligo beads.

Optical tweezers experiments were performed in a custom-made dual-trap optical tweezers instrument modified from the design in Comstock et al (*Whitley et al., 2017*; *Comstock et al., 2011*). In this configuration, a 1064 nm laser is passed through an acousto-optic deflector, with the laser alternating in position between the two traps every 5 μs. The position of the beads relative to the traps was measured using back focal plane interferometry (*Bustamante et al., 2009*). Single tethers were formed in situ inside the chamber by trapping an oligo bead in one trap and an SA bead in the second trap, and bringing in close proximity the two traps to allow the biotin on the right arm of the Y to interact with streptavidin (Streptavidin bead).

The pulling experiments used to obtain the average dwell times in *Figures 1E* and *2B* are non-equilibrium in nature, reflecting an average over many out-of-equilibrium encounters with each

nucleosome type, and are not direct indicators of equilibrium interaction strength alone. In particular, dwell times to rupture in a given trace will depend on the forces reached during a prior rupture event. Complementary probes of DNA-histone interaction strength (see below) verify that many of the peaks in *Figures 1E* and *2B* are due to strong binding interactions.

Due to the nonequilibrium nature of the experiments used to obtain the residence times in *Figures 1E* and *2B*, we caution against the use of such 'pulling' residence times alone to assess interaction strength with high resolution. Data from these pulling experiments is most useful when assessed in conjunction with data from other experimental techniques, such as the equilibrium hopping data in *Figure 3*, and the transcriptional residence times of *Figures 6* and *7*.

### Unzipping at a constant trap separation speed

Once a single tether was confirmed, the trap distance was reset to a value at which the tether force was ~0.4 pN. Unzipping was initiated by moving the two traps apart at a constant speed of 20 nm/s for a total of 875 nm. Under this condition, this pulling speed corresponds to a loading rate of 1.8–2.5 pN/s. Rezipping was conducted at the end of unzipping by bringing the two traps together to the initial trap position at the same speed. The tether was broken manually by increasing the trap distance and calibration was performed as previously described (*Berg-Sørensen and Flyvbjerg, 2004*). Data was acquired at 800 Hz.

### Unzipping at 28 pN constant force

To unzip the construct at constant force, the tether was initially held at ~10 pN and force feedback was turned on to maintain the tether at a constant force of 28 pN. The position and distance between the two beads ware recorded at 800 Hz until the construct was fully unzipped. The force feedback was turned off and the tether was relaxed to ~0.5 pN. For the purpose of aligning the traces and accurately converting nanometer distance to basepairs unzipped, an unzipped trace using constant trap separation speed (as described above) was further obtained from the same tether.

### Partial unzipping up to the proximal dimer region to test nucleosome mobility

Unzipping was performed at constant trap separation speed of 50 nm/s up to the proximal dimer region where the force starts to rise above the baseline of bare DNA construct, but does not reach 30 pN. The partial unzipping was followed by rezipping to the initial trap position. Typically, this results in a trap movement of ~620 nm. After repeating the unzipping-rezipping cycle for 5–10 times, a final unzipping that unzips all the way to the hairpin end (trap movement of 875 nm) was performed to disrupt the whole nucleosome. A bare DNA trace was also collected immediately following this final unzipping.

### Unzipping at constant trap distances to record hopping traces near the proximal dimer interaction region

To capture hopping of the unzipping fork near the proximal dimer interaction region of the nucleosome, the trap was manually moved apart at a small distance increment of 7.1 nm. At each discrete trap position (passive mode), the distance between the two beads was recorded at 2.5 kHz for 10–300 s. Initially, only fast hopping events characteristic of dsDNA unwinding were present. Once the unzipping fork arrived at the proximal dimer interaction of the nucleosome, additional slow transition events, due to histones binding with and dissociating from dsDNA or ssDNA, could be seen. Recording was terminated when the force reached ~23 pN. The tether was then relaxed to ~0.5 pN. For the purpose of aligning the traces and accurately converting nanometer distance to basepairs unzipped, an unzipped trace using constant trap separation speed (as described above) was further obtained from the same tether.

### Construction of the 8 × repeat 'molecular ruler' plasmid

The plasmid that contains a single repeat sequence (pGEM3z-1×repeat) was first cloned by modifying a pGEM3z-T7A1 plasmid (*Gabizon et al., 2018*). The construction of the plasmid with eight tandem repeat sequences (pGEM3z-8×repeat) was carried out by following a published protocol (*Wu et al., 2016*) using BglI, DraIII and EagI restriction sites. This method allows doubling of the

repeat number following each cycle of cloning. To ease isolation and purification of the 8 × repeat DNA for future ligation steps, we removed the internal BsaI site in pGEM3z-8×repeat and introduced two BsaI sites flanking the 8 × repeat region by using an overlap PCR strategy. Briefly, two fragments outside the 8 × repeat region of the plasmid were amplified using oligos ZC12-ZC13 and ZC14-ZC15, respectively, and assembled into one fragment using ZC12-ZC15. The assembled fragment, which is devoid of the internal BsaI site, was digested with SapI/EagI and ligated with SapI/EagI digested 8 × repeat fragment from pGM3z-8×repeat. The resulting plasmid pGM3z-8×repeat-2×BsaI contains the 8 × repeat sequence flanking by two BsaI sites, which are included in oligos ZC12 and ZC15. All plasmids containing repeat sequences were transformed and grew in SURE2 competent cells at 30°C. Large amounts of pGM3z-8×repeat-2×BsaI plasmids were purified from 150 mL of SURE2 cells using Zyppy Plasmid Maxiprep Kit.

## DNA templates for pol II nucleosomal transcription assay

The 8 × repeat DNA with proper overhangs were digested from pGM3z-8×repeat-2×BsaI using BsaI-HF and purified using 8% native PAGE.

The crosslinked DNA (XLink) used to stall Pol II at the end of the template was prepared by annealing ZC16 and ZC17. The annealed oligos were diluted to 1 µM in TE with 20% DMSO and 50 µM trioxsalen, irradiated by 340 nm UV light for 15 min. Extra trioxsalen (10 µM more) was added and the oligos were irradiated for another 15 min. This procedure was repeated to ensure complete crosslinking. The crosslinked oligos were bound to 1 mL HiTrap Q column (GE Healthcare), washed with 4 mL TE buffer, then 4 ml TE buffer + 250 mM NaCl, eluted with TE + 1 M NaCl, and desalted using Amicon Ultra centrifugation filters with Ultracel 3K membrane. The sequences of the crosslinked oligos are:

5' phos-GGTGTACAGAACGCAATGAATT 3'
3' GGACCACATGTCTTGCGTTACTTAA 5'

NPS DNA (308 bp) that contains the 147 bp '601' NPS was amplified from a pGMZ-3z/601 plasmid using oligos ZC18 and ZC19. The NPS DNA was digested with BsaI/DraIII, purified using HiTrap Q column, and ligated to the crosslinked oligo. The ligation product (NPS-Xlink) was purified using 8% native PAGE.

The 2 kb upstream spacer DNA and 1.5 kb biotin handle DNA were amplified from lambda DNA using oligos ZC20-ZC21, ZC22-ZC23, respectively. PCR products were digested with BsaI and purified using 1% agarose gel. Both the 2 kb spacer and 1.5 kb biotin handle DNA were ligated to 5'-CGGT 1 µm polystyrene oligo beads overnight at 16°C using T4 DNA ligase (NEB). The ligated beads were first washed with TE + 0.5 M KCl + 20 µg/mL β-casein, then washed twice with TE + 20 µg/mL β-casein and resuspended in TE + 20 µg/mL β-casein to a concentration of 0.02% (w/v) for 1.5 kb biotin handle, and 0.2% (w/v) for 2 kb spacer DNA. The ligated beads were stored at 4°C until experiments.

## Assembly of yeast pol II stalled complex

Biotinylated yeast Pol II holoenzyme was expressed, purified and biotinylated as previously described (*Kireeva et al., 2003*) and was a generous gift of Prof. Craig Kaplan. The stalled Pol II elongation complex was prepared by a bubble initiation method followed by uridine triphosphate (UTP) starvation (*Hodges et al., 2009*). The sequences for the template DNA strand (TDS), non-template DNA strand (NDS) and short RNA (RNA9) are:
*NDS:*
5'A**GGTCTCAGAAG**ACGCCCGAACAACAGACACAAACACCACGGCCGGCGAGCCAGACACGAC-
CAA**T**TATCTATGTAACTTGCCATATTCAGGATTAT 3'
*RNA9:*
5' GACGCCCGA 3'
*TDS:*
3'T**CCAGAGTCTTC**TGCGGGCTTGTTGTCTGTGTTTGTGGTGCCGGCCGCTCGGTCTGTGCTGG
TT**A**ATAGATACATTGAACGGTATAAGTCCTAATAGTCA-phos 5'

To assemble the stalled complex, TDS was incubated with RNA9, heated to 45°C and cooled down to 20°C at 1 °C/min to form the TDS/RNA9 hybrid. Pol II was added to the hybrid and incubated at room temperature (RT) for 10 min, followed by NDS addition and incubation at 37°C for 15

min. Transcription was initiated by adding ATP/GTP/CTP to a final concentration of 10 µM each and the reaction was incubated at RT for 10 min. If Pol II succeeded in restarting, it will be stalled at the first A site on TDS (bolded and underlined in the sequence above) due to absence of UTP. The relocation of Pol II to the stall site will also expose a BsaI site (underlined above) shielded initially by Pol II and only those complexes in which Pol II succeeded in restarting can be digested and further ligated to the 2 kb upstream spacer. The stalled complex was digested with BsaI-HF at 37°C for 15 min, aliquoted, and stored at −80°C until usage.

## Nucleosome reconstitution on NPS-Xlink template

*Xenopus* WT (xWT) nucleosome was reconstituted by salt-dialysis using NPS-Xlink DNA with *Xenopus laevis* recombinant histone octamer. Human WT (hWT), uH2B and H2A.Z nucleosomes were reconstituted similarly to those used in the single-molecule unzipping assay, except that NPS-Xlink DNA was used. The efficiency of nucleosome reconstitution was assessed by 4% native PAGE. In case where a significant amount of free DNA was present, the nucleosome was further purified by sucrose gradient ultracentrifugation. The nucleosomes were concentrated, supplemented with 0.02 % NP40 and stored at 4°C.

## Optical tweezers assay of pol II transcription through the nucleosome

Transcription was performed in TB50 buffer. NTPs concentration was 0.5 mM each of ATP, CTP, GTP, UTP. The 1.5 kb biotins beads were pre-incubated with 0.5 µM neutravidin for 10 min at room temperature and diluted with TB50. Pol II sample beads were prepared by ligating the 1 µm 2 kb spacer DNA beads, Pol II stalled complex, 8 × repeat DNA and nucleosome loaded on NPS-Xlink (or bare NPS-Xlink DNA) using *E. coli* DNA ligase (NEB) at 16°C for 2 hr. 0.02% of NP40 was also included in the ligation reaction. The overhangs of the various components were optimized such that the ligation occurs at desired orders. The sample beads were diluted with TB50 + 0.02% NP40. The full sequence of the assembled transcription template was available at the end of the document.

To perform the experiment, we first captured a 1.5 kb biotin bead in one trap followed by a Pol II sample bead in the other trap. The two beads were rubbed against each other until a tether is formed. If the tether has expected length, the pair of beads was moved to the experimental position, which is close to the outlet of the NTPs channel. Force feedback was turned on to maintain a constant force of 10 pN and the NTPs channel was opened to start transcription. Data acquisition was started right after force feedback was turned on and terminated once the polymerase reached the end or arrested for more than 300 s without dynamics. To probe the fate of transcribed nucleosomes, force feedback was turned off and the trap distance was reset to a value that gives less than 1 pN force on the tether. The two beads were pulled away from each other by increasing trap distance at a constant speed of 20 nm/s, until the force reaches above 40 pN. From these pulling curves, we rarely detected rips normally associated with nucleosome unwrapping. Trap distance was further increased to break the tether and the beads were calibrated. All transcription data was recorded at 800 Hz.

## In vitro pol II transcription on the 1 × repeat template

The 1 × repeat DNA template was amplified from pGM3z-1×repeat plasmid using oligos ZC24-ZC25, digested with BsaI-HF and purified by agarose gel extraction. To determine the main pause site in the 64 bp repeat sequence, Pol II stalled complex was radioactively labeled with [α-$^{32}$P]-ATP during initial pulsing. The stalled complex was loaded on streptavidin-coated magnetic beads. The beads were washed with TB130 (20 mM Tris, pH = 8.0, 130 mM KCl, 5 mM MgCl$_2$, 10 mM DTT, 20 µg/mL BSA) and ligated to the 1 × repeat DNA template using T4 DNA ligase for 1 hr at RT. Transcription was chased by adding 40 µM NTPs mix (ATP, UTP, CTP, GTP, final concentration of 40 µM each) to the stalled complex beads, and terminated by adding 2 × urea stop buffer (8 M urea, 50 mM EDTA) at 10, 20, 40, 60, 120, 300, 600 and 900 s. In parallel, transcription was chased by adding 40 µM NTPs mix together with 50 µM of each type of 3'-deoxynucleotide RNA chain terminators (3'dATP, 3'dCTP, 3'dGTP, 3'dUTP, TriLink Biotechnologies). The reactions were allowed to proceed at room temperature for 10 min before terminated by adding the 2 × urea stop buffer. Samples were extracted with Phenol: Chloroform: Isoamyl alcohol (1:0.9:0.01), precipitated with ethanol and

dissolved in 2 × formamide sample buffer (95% formamide, 5 mM EDTA, pH 8.0, with bromophenol blue and xylene cyanol). RNA was resolved on 12% denaturing PAGE, dried and exposed to a phosphorimager screen. Images were captured on the Typhoon imager (GE Healthcare) and processed by ImageJ.

## Mechanical model for pol II transcription through the nucleosome

### Pol II dynamics

Our model for Pol II dynamics is illustrated in *Figure 7B*. In this model, Pol II takes forward main pathway steps by one base pair at a rate $k_0$ or can enter a backtracked pathway by stepping back one base pair at rate $k_{b1}$. Once backtracked, Pol II takes steps one base pair forward at rate $k_{fn}$ and steps one base pair backward at rate $k_{bn}$. Stepping forward from the first backtracked state returns Pol II to the main pathway.

Transition rates depend on force f, with main pathway and backtracking step rates given by

$$k_0(f) = k_0^0 e^{\delta_0 \ell f/(k_B T)},$$

$$k_{fn} = k_{fn}^0 e^{\delta_0 \ell f/(k_B T)} \text{ and } k_{bn} = k_{bn}^0 e^{-(1-\delta_{fb})\ell f/(k_B T)}$$

$k_0^0$, $k_{fn}^0$, and $k_{bn}^0$ are the zero force rate constants. $\delta_0$ and $\delta_{fb}$ are splitting factors, representing the transition state location. $\ell = 0.34$ nm is the step size, the length of one DNA base pair. $k_B T = 4.11$ pN·nm is the thermal energy at room temperature.

Our model is adapted from the Pol II dynamics model and parameterization of *Dangkulwanich et al. (2013)*. Dangkulwanich models Pol II forward stepping as three stages, with the first two reversible, and the third effectively irreversible. Our experimental condition of high nucleotide concentration leads to a nearly instantaneous second transition, and we combine the two remaining transitions into a single irreversible transition with rate $k_0$. The zero-force forward rates of the two remaining stages in Dangkulwanich are 88 s$^{-1}$ and 35 s$^{-1}$, combined into $k_0^0 = 25$ s$^{-1}$. The rate of initial backstepping, $k_{b1}$, is only from the first of the three main pathway states in Dangkulwanich. Accordingly, we weight this zero-force initial backtracking rate, 6.9 s$^{-1}$, by the probability of being in the main pathway state eligible for backtracking, $k_{b1}^0 = (35/66) \cdot 6.9$ s$^{-1}$. Backtracking is restricted to a maximum of three base pairs, such that $k_{bn} = 0$ for $n \geq 4$. The remaining parameters are $k_{fn}^0 = 1.3$ s$^{-1}$ for all $n$, $k_{bn}^0 = 1.3$ s$^{-1}$ for $1 \leq n \leq 3$, $\delta_0 = 0.64$, and $\delta_{fb} = 0.5$, taken directly from Dangkulwanich.

### Nucleosome effect on polymerase kinetics

The model above describes transitions of the polymerase on DNA, but does not incorporate the effect of the nucleosome, which is expected to hinder forward stepping. We adapt a previous model (*Koslover and Spakowitz, 2012*) to describe the DNA polymerase-nucleosome system on a two-dimensional energy landscape ($E_{j,w}$). The first dimension ($j$) is the position of the polymerase and the second ($w$) is the number of DNA base pairs unwrapped from the nucleosome. This energy landscape incorporates the mechanics of the DNA, polymerase, and nucleosome interaction (namely, steric exclusion between polymerase and nucleosome and bending of the unwrapped DNA) as described in the section below.

For a given length of unwrapped DNA, there is a change in energy associated with the polymerase stepping forward,

$$\Delta E_{j,w} = E_{j+1,w} - E_{j,w},$$

which modulates the rate of that step according to

$$k_0^{j,w}(f) = k_0^0 \exp[\delta_0(\ell f - \Delta E_{j,w})/(k_B T)].$$

This assumes that the step forward involves a transition state at fractional position $\delta_0$ and that the energy landscape is linear between positions $j$ and $j+1$.

We assume that the wrapping and unwrapping of DNA from the nucleosome is much faster than the polymerase stepping kinetics. In this case, the system is equilibrated along the $w$ dimension, and

the overall stepping rate for the polymerase can be described as a weighted average over all the stepping rates:

$$k_0^{(j,\text{eff})} = \left(k_0^0 e^{\frac{\delta_0 \ell f}{k_B T}}\right) \frac{\sum_w e^{-E_{jw}} e^{-\delta_0 \Delta E_{jw}}}{\sum_w e^{-E_{jw}}}$$

An analogous calculation is done for the forward and backward stepping rates in the backtracked state:

$$k_{\text{bn}}^{(j,\text{eff})} = \left(k_{\text{bn}}^0 e^{\frac{(1-\delta_{\text{fb}}) \ell f}{k_B T}}\right) \frac{\sum_w e^{-E_{jw}} e^{(1-\delta_{\text{fb}}) \Delta E_{j-1,w}}}{\sum_w e^{-E_{jw}}}$$

$$k_{\text{fn}}^{(j,\text{eff})} = \left(k_{\text{fn}}^0 e^{\frac{\delta_{\text{fb}} \ell f}{k_B T}}\right) \frac{\sum_w e^{-E_{jw}} e^{-\delta_{\text{fb}} \Delta E_{j,w}}}{\sum_w e^{-E_{jw}}}$$

(all the energies in the above are expressed in units of $k_B T$). Overall, the presence of the nucleosome modifies the polymerase kinetics by making it much slower to step forward if doing so would require a substantial increase in energy associated with bending of the linker DNA ahead of the polymerase.

## Energy landscape for polymerase–nucleosome system

The free energy $E_{jw}$ is defined by the location of the polymerase at basepair $j$ (relative to the start of the nucleosome) and the number of DNA base pairs unwrapped from the nucleosome, $w$.

$$E_{jw} = E_N^{(L)} + E_{\text{int}}$$

The first term $E_N^{(L)}$ is the conformation energy for the DNA linker $N$ base pairs in length between Pol II and the nucleosome (*Koslover and Spakowitz, 2012*). We use a highly simplified mechanical model for this system, where the histone core of the nucleosome is treated as a steric sphere of radius $R_{\text{nuc}}$ = 3.2 nm and Pol II is treated as a steric sphere of radius $R_{\text{pol}}$ = 7 nm. The DNA is modeled as a wormlike chain with 35.4 nm persistence length, that must stretch from the center of the polymerase to positions along a spiral wrapped around the nucleosome (*Figure 7A*). For a given length of DNA unwrapped ahead of the polymerase ($\ell N$), the bending energy is calculated by optimizing the wormlike chain configuration subject to the constraint that the steric spheres for polymerase and nucleosome may not overlap. If very little DNA is unwrapped ahead of the polymerase, the linker is short and must bend tightly to avoid steric overlap (leading to high energies). If more of the DNA is unwrapped, the linker may not need to bend at all ($E_N^{(L)}$=0 for lengths above approximately 30 bp).

The second contribution $E_{\text{int}}$ is the energy of DNA interaction with the nucleosome. This includes unfavorable bending of the DNA around the nucleosome and favorable DNA-nucleosome binding interactions. $N_{\text{tot}}$ = 147 base pairs can bind to the nucleosome, and each can have a different interaction energy. For $w$ DNA base pairs unwrapped from the nucleosome

$$E_{\text{int}} = \sum_{i=w+1}^{N_{\text{tot}}} \phi_i,$$

where $\varphi_i$ is the interaction energy of $i$'th base pair with the nucleosome.

## Determining dwell times and fitting

With the quantitative model of polymerase dynamics, we can determine mean dwell times. We analytically determine the mean time for the polymerase to reach the $n+1$'th state after first reaching the $n$'th state (*Koslover and Spakowitz, 2012*).

This model assumes the binding/unbinding of the DNA ahead of the polymerase is always equilibrated as the polymerase steps backward and forward. This is a reasonable assumption, given the rapid equilibration time for DNA unwrapping, but only up to the point when the DNA fully unwraps from the nucleosome. Our model neglects the additional entropic contributions of DNA and polymerase separating completely in solution and cannot properly predict the dwell times at the very end of the polymerase transcribing through the nucleosome.

Using the lsqcurvefit routine in Matlab, we fit the DNA-nucleosome interaction energies $\varphi_i$ to match the quantitative model mean dwell times to the experimental mean dwell times, smoothed by taking the local average over a 3 bp span. As shown in *Figure 7D*, we only include experimental mean dwell time where the polymerase is positioned within the nucleosomal binding sequence ($j \geq 0$). Prior to these base pairs, we use a mean dwell time $(k_0^0)^{-1} = 0.04$ s.

## Data analysis

### Single-molecule unzipping data analysis

Using the calibration data, we calculated the complete force-extension curves for each tether. The analysis consisted of the following steps:

First, the relaxation of the fully unzipped construct (that is, after the complete unzipping of the construct and before rezipping of dsDNA has begun, corresponding to a force range of ~20–40 pN) was fit to a model in which 1850 bp of dsDNA are described as a worm-like chain with a persistence length of 35.4 nm, a stretch modulus of 1020 pN and a contour of 0.34 nm/bp (*Bouchiat et al., 1999*; *Bustamante et al., 1994*), and 872 bases of ssDNA are described using an extensible freely jointed chain with a contour length of 0.59 nm/base (*Mills et al., 1999*; *Smith et al., 1992*). The dsDNA parameters were estimated by analyzing the pulling curves of 4.7 kb dsDNA molecules. The other parameters (stretch modulus and Kuhn lengths for the ssDNA and an offset of the extension to account for bead size variation) were fit, resulting in a Kuhn length of 1.45 ± 0.02 nm, a stretch modulus of 975 ± 61 pN and an offset of 29 ± 2 nm (errors are 95% confidence intervals over all traces, N = 234). These values are close to previously published values (*Smith et al., 1996*). Using these parameters, we calculated the number of unzipped base pairs at all positions along the pulling trace.

Second, we performed a minor adjustment on the extension to align the two NPS repeats on the bare DNA template. In principle, identical positions in the two NPS repeats should be 197 bp apart in distance, and they are expected to behave identically in the trace (same force-extension signatures). However, the calculated distance obtained initially is typically different from this value of 197 bp. At this point, we rescaled the data along the x-axis (number of unzipped base pairs) to maintain 436 unzipped base pairs at the end of the unzipping curve and a distance of 197 base pairs between identical positions on the two NPS sequences. To find the correct scaling factor, we rescaled the data using a range of scaling factors (from 170 to 197) using the following equation:

$$N_{rescaled} = 436 - \frac{197}{factor} \times (436 - N_{not_{rescaled}})$$

For each factor, we binned the data points in 0.5 bp window and calculated the force-weighted residence histogram along the sequence. We then calculated the correlation between the histogram at positions along the first NPS and the histogram at positions along the second NPS:

$$Correlation = \sum_{first\_copy\_i} Res(first\_copy\_i) \times Res(first\_copy\_i + 197)$$

The factor giving the maximum correlation was selected, and the data was finally rescaled using this factor. Using this approach, we generated a mean residence histogram of the first NPS from all bare DNA unzipping traces. The rescaling factors were typically between 180 to 190. The requirement for rescaling to satisfy the periodicity may result from bead size variations or deviations from the models used to describe the pulling traces.

Third, once an aligned mean residence histogram of the first NPS was obtained from unzipping traces of bare DNA, a slightly modified operation was performed on unzipping traces of the nucleosome datasets. Again, the relaxation after complete unzipping was fit and the number of unzipped base pairs were calculated, and again rescaled using a range of rescaling factors. This time, the correlation between the residence time histogram of the first NPS in the nucleosome traces and the mean residence histogram of the first NPS obtained in the previous step was calculated and maximized. The rescaling factors for nucleosome data had the same range as for the bare DNA data.

### Residence time analysis of unzipping traces

After obtaining the fitting parameters for both dsDNA and ssDNA, bead-to-bead distances of the unzipping traces were converted to unzipped basepairs. The unzipped basepairs of the traces were then aligned, scaled and normalized to the beginning of the second NPS by subtracting 248 bp (the second NPS begins at 249 bp of the Y stem region). For traces obtained at constant trap separation speed (20 nm/s), a force weighted residence time (RT) between each bp was calculated by summing the forces of all data points between two consecutive unzipped basepairs (*Figure 1E*). Therefore, long residence time (i.e. more data points) while under higher force within a particular bp would result in a high force-weighted RT in this analysis. The force-weighted RT accounts for force differences along the unzipping trace and serves as a proxy of the strength of histone-DNA interactions of the nucleosome. For constant force unzipping traces, residence time at each bp was calculated by counting intervening data points N. Because data frequency is 800 Hz, RT therefore equals to N/800 (*Figure 2C*). RT histograms are plotted as mean values from all traces.

### Analysis of the number of unzipping transitions in unzipping traces

A transition in the unzipping trace is defined as a peak in the residence time histogram that is above a certain threshold. For each unzipping trace obtained at constant trap separation speed, we identified transition events by looking for maxima in the RT histogram and manually applying a threshold to avoid too many transitions (rips) from just bare DNA. The chosen threshold cannot be too high, as the RT for H2A.Z unzipping traces generally have more peaks but lower amplitude for each peak. This analysis (*Figure 1—figure supplement 1F*) revealed that on average, H2A.Z nucleosome unzipping traces have at least one more ripping transition than those of WT nucleosomes.

### Analysis of the partial unzipping data to test mobility

The final unzipping trace or the bare DNA unzipping trace was used to fit the WLC model to obtain the elasticity, offset and scaling parameters of a particular tether. These parameters are applied to previous partial unzips from the same tether. All traces for a particular tether are plotted together without any further alignments. Note, for both WT and H2A.Z nucleosomes, the initial force rise always occurs at the same position without lateral shifts. During the force rise at the proximal dimer region, the unzipping fork randomly dwells at nearby locations (~5 bp away), consistent with nucleosome hopping in this region.

### Analysis of hopping (equilibrium) data at constant trap positions

To explore steady-state behavior of DNA on the nucleosome, the trap separation was held fixed such that the DNA experiences wrapping and unwrapping fluctuations in the proximal dimer region, 'hopping' on and off the nucleosome. A trace of force-extension pairs is measured at each trap separation (*Figure 3—figure supplement 1F*), followed by a final unzipping and relaxation trace at constant trap velocity. The following subsections describe our analysis methods for extracting from this data an underlying energy landscape for DNA base pairing energies and the energies of interaction with the nucleosome.

### Calculation of unzipping energies from force-extension traces

Because the pulling and extension curves for bare DNA overlap closely with no hysteresis (*Figure 1B*), we assume this process is at equilibrium. The energy associated with unzipping each basepair can then be computed from the work done by the pulling force during unzipping, with a correction for the work required to extend the newly unzipped bases.

To start, we find the fractional extension of dsDNA worm-like chains and ssDNA freely jointed chains at a given force, $z_{ds}(F)$ and $z_{ss}(F)$, respectively (*Wang et al., 1997*). The length of ssDNA $L_{ss}$ between the two dsDNA handles of length $L_{ds}$ for each force-extension pair $(F,s)$ is then given by

$$L_{ss} = \frac{s - 2z_{ds}(F)L_{ds}}{z_{ss}(F)}$$

The number of base pairs unzipped is

$$N_{\text{unzip}} = \frac{L_{ss}}{2l_{ss}},$$

where $\ell_{ss}$ is the ssDNA length per base pair, with a factor of two because twice the base pair length of ssDNA is obtained when unzipping one base pair of dsDNA. Each number of base pairs unzipped $N_{\text{unzip}}$ can now correspond to a specific force $F$, length of ssDNA $L_{ss}$, and fractional extension of ssDNA $z_{ss}$. The unzipping energy of each base pair of ssDNA is the overall work to extend the two newly unzipped bases minus the work required to stretch those bases to the observed extension.

$$\Delta E_{\text{unzip}}(N_{\text{unzip}}) = 2Fz_{ss}l_{ss} - \int_0^{2l_{ss}z_{ss}} F_{\text{FJC}(x)}dx.$$

## Alignment of force-extension data

Because the bead radius cannot be known precisely, individual data collection runs are shifted arbitrarily along the extension axis. We use the final complete pulling curve to account for this shift. The pulling curves for each experimental run with bare DNA are first mutually aligned (*Figure 3—figure supplement 1C*) and the average trace is used to calculate the absolute shift along the extension axis.

Specifically, we calculate the unzipping energy for each basepair as described in the previous section. The two copies of the NPS give rise to duplicate features in the base-pair interaction energy landscape, whose separation depends on the absolute values of the end-to-end extension input into the calculation. We therefore shift the averaged force-extension curve along the x axis in such a way that the duplicate energy features are separated by precisely 197 bp (*Figure 3—figure supplement 1D*). The same shift is assumed for the equilibrated hopping data obtained for each individual DNA molecule prior to the corresponding pulling trace. No scaling of the x-axis is done in this analysis.

In our calculations we used dsDNA persistence length of 35.4 nm, dsDNA stretching modulus 1020 pN, 0.34 nm contour length per base pair, ssDNA segment length 1.03 nm, ssDNA stretching modulus 1000 pN, 0.59 nm contour length per base pair. The ssDNA parameters were obtained by fitting the final region of the averaged pulling curve for bare DNA traces, where the hairpin has been completely unzipped. The calculated force-extension relation for a molecule with a 434 bp unzippable region, terminated with a 4-base hairpin, and connected to two dsDNA handles (1848 bp), given the fitted unzipping energies, is shown in *Figure 3—figure supplement 1E*.

Pulling traces with bound nucleosomes present are aligned to the averaged pulling trace for bare DNA based on the force-extension curve features prior to reaching the second NPS (specifically, extensions below 870 nm are used for alignment).

## Extracting DNA-nucleosome interaction energies from equilibrated hopping data

For each trap separation, the number of base pairs unzipped ($N$) is obtained for each force-extension pair, populating a distribution in $N$ (*Figure 3B*). Assuming the system is in thermodynamic equilibrium, the probability for each number of base pairs unzipped is converted to a relative energy for each number of base pairs unzipped. Subtracting the energy of DNA stretching and the energy for the off-center beads in the optical traps gives the cumulative relative energy to unzip the given number of base pairs. The difference in this cumulative relative energy between consecutive base pairs is the energy to unzip each base pair. The various fixed trap separations provide overlapping ranges for the energy of unzipping for each basepair (*Figure 3—figure supplement 1G*), and the average value from all trap separations that span a particular value of N is used for further analysis.

To find the energy of the DNA-nucleosome interactions, the unzipping energy for bare DNA (no nucleosome present) is subtracted from the unzipping energy for a system with a nucleosome present (WT, H2A.Z, and uH2B).

Given the extracted energies of unzipping bare DNA and peeling DNA off the nucleosome for the region accessed by the equilibrium hopping data, we can calculate the predicted force extension relation in an equilibrium pulling curve (*Figure 3—figure supplement 1H*). We note that the observed forces in the nucleosome-bound region during the constant velocity pulling traces are substantially higher, emphasizing that these traces are obtained out of equilibrium.

## Nucleosome transcription data analysis

The alignment of the 'molecular ruler', data analysis on pausing, backtracking and residence time of Pol II was performed essentially the same as recently described (*Gabizon et al., 2018*). Briefly, for each trace the region expected to contain the repeats (8 × 64 bp) was aligned to find the physical length of the repeat in nanometers, and the aligned traces were aligned between themselves and to the known pause sites discovered by biochemical studies (described below). The pause site within each repeat is located at the 59th nucleotide (T) of the 64 bp DNA and the periodicity of the physical length of each repeat is found to be 21.1 nm at 10pN force. The position of the polymerase along the nucleosome was obtained by extrapolating the position from the aligned repeat region. To plot the transcribed distance (bp) of the leading edge of Pol II relative to NPS, we applied an offset of 16 bp to account for the footprint of Pol II.

The crossing time is calculated as the total duration of the leading edge of Pol II crossing the entire 147 bp NPS region. Only traces that reached the stall site are included in crossing time analysis.

Example traces of Pol II hopping at certain regions (*Figure 5—figure supplement 1*) were analyzed with a classic Hidden Markov Model (HMM) by fitting to two (for hWT or H2A.Z) or three states (for uH2B).

Probability of arresting is calculated as the percentage of traces that entered NPS but did not reach the stall site, while probability of crossing is the percentage of traces that successfully reached the stall site. Typically, we considered a trace that paused 300 s or longer without any associated dynamics to be arrested. For arrested traces, percentages of traces that arrested before or after the dyad are also calculated based on their arrest position.

Pause-free velocities (bp/s) of Pol II before, inside and after NPS are estimated by calculating the inverse of the median residence time (s/bp) at distinct sites. To account for sequence bias, the three fastest sites (lowest median residence time) are chosen from each sampling range. For regions before or after NPS, sites up to 100 bp away from the NPS region are sampled.

## Full sequence of the unzipping template

The sequence below shows the stem of the Y structure that contains two consecutive pieces of NPS DNA (bold and italic). The red sequence is the stem of the end hairpin (four bases of the loop not shown here). The 147 bp core '601' NPS is underlined in each segment. The upstream DNA is bridged to the two arms of the Y. The DNA in between are ligation sites. The full length of ssDNA after complete unzipping will be 872 bases (including extra bases from the loop of the hairpin).

(arms)...TTTTGACTACTGACGCGGACATTCAGGA***GATGGACCCTATACGCGGCCGCCCTGGA-GAATCCCGGTGCCGAGGCCGCTCAATTGGTCGTAGACAGCTCTAGCACCGCTTAAACGCACG TACGCGCTGTCCCCCGCGTTTTAACCGCCAAGGGGATTACTCCCTAGTCTCCAGGCACGTG TCAGATATATACATCCTGT*GCATGTATTGAACAGCGACCTTG*CAAC*GATGGACCCTA TACGCGGCCGCCCTGGAGAATCCCGGTGCCGAGGCCGCTCAATTGGTCGTAGACAGCTCTAG-CACCGCTTAAACGCACGTACGCGCTGTCCCCCGCGTTTTAACCGCCAAGGGGATTACTCCCTAG TCTCCAGGCACGTGTCAGATATATACATCCTGT*GCATGTATTGAACAGCGACCTTG*** CACCCTCCACTCTAGA

## Full sequence of the Pol II transcription template

Pol II loading sequence (NDS/TDS), 8 × repeat DNA, core '601' NPS and crosslinked DNA (Xlink) are in blue, bold, italic and red, respectively. The transcription starvation site (T in NDS) is bolded and underlined. The NPS-Xlink DNA used for octamer loading is underlined.

AGGTCTCAGAAGACGCCCGAACAACAGACACAAACACCACGGCCGGCGAGCCAGACAC-GACCAA**T**TATCTATGTAACTTGCCATATTCAGGATTATCAGTAGCGGAAGAGCGAGCTCGG TACCCGATCCAGATCCCGAACGCCTATCTTAAAGTTTAAACATAAAGACCAGACCTAAAGACCA-GACCTAAAGACACTACATAAAGACCAGACCTAAAGACGCCTTGTTGTTAGCCATAAAGTGA TAACCTTTAATCATTGTCTTTATTAATACAACTtACTATAAGaAGAGACAACTTAAAGAGAC TTAAAAGATTAATTTAAAATTTATCAAAAAGAGTATTGACTTAAAGTCTAACCTATAGGATACTTA-CAGCCATCGAGAGGGACACGGGGAAACACCACCAGCCT**CCCGGGCTCACCATCATCCTGAC TAGTCTTTCAGGCGATGTGTGCTGGAAAGATCTTATGTCACCCCGGGCTCACCATCATCCTGAC TAGTCTTTCAGGCGATGTGTGCTGGAAAGATCTTATGTCACCCCGGGCTCACCATCATCCTGAC**

**TAGTCTTTCAGGCGATGTGTGCTGGAAAGATCTTATGTCACCCCGGGCTCACCATCATCCTGAC**
**TAGTCTTTCAGGCGATGTGTGCTGGAAAGATCTTATGTCACCCCGGGCTCACCATCATCCTGAC**
**TAGTCTTTCAGGCGATGTGTGCTGGAAAGATCTTATGTCACCCCGGGCTCACCATCATCCTGAC**
**TAGTCTTTCAGGCGATGTGTGCTGGAAAGATCTTATGTCACCCCGGGCTCACCATCATCCTGAC**
**TAGTCTTTCAGGCGATGTGTGCTGGAAAGATCTTATGTCACCCCGGGCTCACCATCATCCTGAC**
**TAGTCTTTCAGGCGATGTGTGCTGGAAAGATCTTATGTCAC**CCCGTGGA
TCCGCCGGCCG<u>CAACGATGGACCCTATACGCGGCCGCCCTGGAGAATCCCGG</u>
<u>*TGCCGAGGCCGCTCAATTGGTCGTAGACAGCTCTAGCACCGCTTAAACGCACGTACGCGCTG*</u>
<u>*TCCCCCGCGTTTTAACCGCCAAGGGGATTACTCCCTAGTCTCCAGGCACGTGTCAGATATATACA*</u>
<u>*TCCTGT*GCATGTATTGAACAGCGACCTTGCCGGTGCCAGTCGGATAGTGTTCCGAGCTCCCACTC</u>
<u>TAGAGGATCCCCGGGTACCGAGCTCGAATTCGCCCTATAGTGAGTCGTATTACAATTCAC</u>
<u>TGGCCGTCGCACCCT</u>GGTGTACAGAACGCAATGAATT

### Full sequence of the 1 × repeat template for in vitro transcription

The 1 × 64 bp repeat sequence is in bold. CAACGCCT**CCCGGGCTCACCATCATCCTGACTAGTC**
**TTTCAGGCGATGTGTGCTGGAAAGATCTTATGTCAC**CCCGTGGATCCGCCGGCCGTCATCACCA
TCATCCTGACTAGAGTCCTTGGCGAACCGGTGTTTGACGTCCAGGAATGTCAAATCCGTGGCG
TGACCTATTCCGCACCGCTGCG

### Data and code availability

Raw data will be made available from the Dryad Digital Repository: https://doi.org/10.5061/dryad.8sb6h8n.

MATLAB scripts have been deposited in GitHub (*Chen, 2019*; copy archived at https://github.com/elifesciences-publications/dataprocessDNAunzipping).

## Acknowledgements

We thank Guillermo Chacaltana and Robert Sosa for technical assistance, and all members of Bustamante laboratory for critical discussion. This work was supported by Howard Hughes Medical Institute, NIH Grants R01GM032543 and R01GM071552 to CB, R01GM098401 to TY, R01GM097260 to CK, the US Department of Energy Office of Basic Energy Sciences Nanomachine Program under Contract DE-AC02-05CH11231 to CB, a grant from Alfred P Sloan Foundation to EFK (FG-2018–10394), and a Burroughs Welcome Fund Collaborative Research Travel Grant to TY; CB is a Howard Hughes Medical Institute Investigator.

## Additional information

### Funding

| Funder | Grant reference number | Author |
| --- | --- | --- |
| National Institute of General Medical Sciences | R01GM032543 | Carlos Bustamante |
| National Institute of General Medical Sciences | R01GM071552 | Carlos Bustamante |
| National Institute of General Medical Sciences | R01GM098401 | Tingting Yao |
| National Institute of General Medical Sciences | R01GM097260 | Craig D Kaplan |
| Basic Energy Sciences | Nanomachine Program under Contract DE-AC02-05CH11231 | Carlos Bustamante |
| Alfred P. Sloan Foundation | FG-2018-10394 | Elena F Koslover |
| Burroughs Wellcome Fund | Collaborative Research Travel Grant | Tingting Yao |

| Howard Hughes Medical Institute | Carlos Bustamante |
|---|---|

The funders had no role in study design, data collection and interpretation, or the decision to submit the work for publication.

## Author contributions

Zhijie Chen, Conceptualization, Resources, Data curation, Formal analysis, Investigation, Methodology, Writing—original draft, Project administration, Writing—review and editing; Ronen Gabizon, Conceptualization, Resources, Data curation, Formal analysis, Investigation, Methodology, Writing—review and editing; Aidan I Brown, Software, Formal analysis, Investigation, Writing—review and editing, Analyzed unzipping data and developed the mechanical model; Antony Lee, Software, Formal analysis, Writing—review and editing, Cloned the initial 1 x repeat template, Developed the algorithm for analyzing transcription data; Aixin Song, Resources, Purified human histones and assembled human octamers; César Díaz-Celis, Resources, Methodology, Assembled *Xenopus* nucleosomes; Craig D Kaplan, Resources, Funding acquisition, Provided biotinylated yeast Pol II; Elena F Koslover, Software, Formal analysis, Funding acquisition, Investigation, Writing—review and editing, Analyzed unzipping data and developed the mechanical model; Tingting Yao, Conceptualization, Resources, Data curation, Supervision, Funding acquisition, Investigation, Methodology, Project administration, Writing—review and editing; Carlos Bustamante, Conceptualization, Supervision, Funding acquisition, Project administration, Writing—review and editing

## Author ORCIDs

Zhijie Chen (iD) https://orcid.org/0000-0003-1376-5750
Ronen Gabizon (iD) https://orcid.org/0000-0002-3626-5073
Aidan I Brown (iD) https://orcid.org/0000-0002-6600-8289
Antony Lee (iD) https://orcid.org/0000-0003-2193-5369
Aixin Song (iD) https://orcid.org/0000-0002-4377-7528
César Díaz-Celis (iD) https://orcid.org/0000-0002-6659-8434
Craig D Kaplan (iD) https://orcid.org/0000-0002-7518-695X
Elena F Koslover (iD) https://orcid.org/0000-0003-4139-9209
Tingting Yao (iD) https://orcid.org/0000-0003-4101-9691
Carlos Bustamante (iD) https://orcid.org/0000-0002-2970-0073

## Decision letter and Author response

Decision letter https://doi.org/10.7554/eLife.48281.034
Author response https://doi.org/10.7554/eLife.48281.035

# Additional files

## Supplementary files

• Supplementary file 1. Oligos used in this study.
DOI: https://doi.org/10.7554/eLife.48281.028
• Supplementary file 2. Pol II backtrack analysis.
DOI: https://doi.org/10.7554/eLife.48281.029
• Transparent reporting form
DOI: https://doi.org/10.7554/eLife.48281.030

## Data availability

Matlab scripts for processing unzipping curves and hopping data have been deposited in GitHub at https://github.com/lenafabr/dataprocessDNAunzipping (copy archived at https://github.com/elifes-ciences-publications/dataprocessDNAunzipping). Raw data are available from Dryad https://doi.org/10.5061/dryad.8sb6h8n. Further information and requests for resources and reagents should be directed to and will be fulfilled by the Lead Contact, Carlos J Bustamante (carlosb@berkeley.edu).

The following dataset was generated:

| Author(s) | Year | Dataset title | Dataset URL | Database and Identifier |
|---|---|---|---|---|
| Chen Z, Gabizon R, Brown A, Lee A, Song A, Diaz-Celis C, Kaplan C, Koslover E, Yao T, Bustamante C | 2019 | Data from: High-resolution and high-accuracy topographic and transcriptional maps of the nucleosome barrier | https://doi.org/10.5061/dryad.8sb6h8n | Dryad Digital Repository, 10.5061/dryad.8sb6h8n |

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
