## [Decision Letter]

Thank you for submitting your article "High-resolution and high-accuracy topographic and transcriptional maps of the nucleosome barrier" for consideration by *eLife*. Your article has been reviewed by two peer reviewers, and the evaluation has been overseen by a Reviewing Editor and Jessica Tyler as the Senior Editor. The reviewers have opted to remain anonymous.

The reviewers have discussed the reviews with one another and the Reviewing Editor has drafted this decision to help you prepare a revised submission.

Summary:

Chen et al. address the question of how Pol II transcribes through the nucleosome, what does its energy landscape look like, and how is it modified by epigenetic markers. They combine two single-molecule assays, DNA unzipping and transcription translocation, to probe the sequence-dependence of nucleosome unwrapping and transcription elongating. An impressive combination of technical innovations allowed them to achieve single base pair accuracy in both assays. Additional unzipping barriers due to the nucleosome were identified, predominantly at the proximal side. Histone variants H2A.Z and uH2B yielded different, larger barriers. A significant increase in pausing during transcription was observed when the substrate contained a nucleosome. Importantly, the location and amplitude of the pauses could be related to the unzipping energy landscape, and using a sophisticated model, a dwell-time distribution as well as an energy landscape for transcription along the nucleosome could be reconstructed. Though both H2A.Z and uH2B nucleosomes slowed down transcription relative to wild type nucleosomes, they introduce different modifications to the energy landscape: H2A.Z introduces a broader distribution of barriers to transcription whereas uH2B increases the height of some of the barriers.

This manuscript describes a major advance in the field. Technically, the accuracy improvements of both assays are significant and allow for a much more detailed interpretation of the data than was reported before. The reported hopping behavior of nucleosomal unwrapping for example has not been observed with such precision.

The molecular mechanisms that are extracted, including detailed energy landscapes, are new and provide the field with a detailed reference for other processes that involve nucleosome transactions.

The impact on our understanding of the biology is significant: the detailed insight reported here provides mechanistic, rather than correlational or kinetic understanding of epigenetic modifications.

Major point:

Regarding the analysis method used to construct the topographic maps (e.g., Figure 1): They point out that the dwell time for each force-vs.-extension contour reflects the strength of the local histone-DNA interaction. As they also point out, what really matters is the force where the transition occurs. The higher the transition force, yes, the longer the dwell time. But also the dwell time depends on where the force began, which depends on both the previous transition force and the amount of DNA released (gives the size of the resultant force drop). All the transitions are in a similar force range and the amount of released DNA vary by factors of 2-4. So it seems that the reported dwell times for a given rupture are just as dependent on how far the previous rupture dropped in force as they are on the magnitude of their own rupture force. This would create a strong convolution and severely scramble the conclusions drawn from the map. For example, in Figure 1C of hWT unzipping, we see two distinct ruptures at very high force near 30 pN. The first rupture has a very long dwell time because the preceding rupture occurred at only 22 pN, while the second has a very short dwell despite needing also 30 pN to rupture because the preceding rupture also occurred at 30 pN. So while these two transitions which required similarly high force to rupture get scored by this method very differently, with the second appearing as a very weak interaction. It seems they are aware of the possibility of such an underestimation of the interaction. But the same effects also overestimate the first rupture. Why can they not measure the distribution of rupture forces (similar to protein folding force spectroscopy methods) directly? Otherwise, they should validate their analysis method on e.g., a long well modeled hairpin or provide some better justification that the conclusions from this method are not seriously flawed.

We have attached the full reviews below, in case these may be of additional assistance to you.

*Reviewer #1:*

Chen et al. and coworkers address the question of how RNA polymerase (Pol II) transcribes through the nucleosome, what does the energy landscape for transcribing through the nucleosome look like, and how is it modified by epigenetic markers. They combine two state of the art single-molecule essays, i.e. DNA unzipping and transcription translocation, to probe the sequence-dependence of nucleosome unwrapping and transcription elongating.

The unzipping essay was improved by aligning the unzipping traces, yielding highly reproducible sequence-dependent features with single base pair accuracy. When applied to nucleosomes, additional unzipping barriers were identified, predominantly at the proximal site of the nucleosomes. Histone variants H2A.Z and uH2B yielded different, larger barriers.

The translocation essay was improved by linking a DNA tether to the Pol II. Using an array of known transcription pause sites in the DNA substrate, translocation traces were aligned, and again single-base pair accuracy was achieved. A significant increase in pausing was observed when the substrate contained a nucleosome.

Importantly, the location and amplitude of the pauses could be related to the unzipping energy landscape, and using a sophisticated model, a dwell-time distribution as well as an energy landscape for transcription along the nucleosome could be reconstructed. Though both H2A.Z and uH2B nucleosomes slowed down transcription relative to wild type nucleosomes, they introduce different modifications to the energy landscape: the first introduces a broader distribution of barriers to transcription. The second increases the height of several barriers.

This manuscript describes a major advancement in the field. Technically, the accuracy improvements of both essays are significant and allow for a much more detailed interpretation of the data than was reported before. The reported hopping behavior of nucleosomal unwrapping for example has not been observed with such precision. The creative design and assembly of the substrates may be a small advance in biochemical engineering, but has a major impact on the accuracy of the results.

The molecular mechanisms that are extracted, including detailed energy landscapes, are new and provide the field with a detailed reference for other processes that involve nucleosome transactions.

The impact on our understanding of the biology is significant: the detailed insight reported here represents one of the few studies that provide mechanistic, rather than correlational or kinetic understanding of epigenetic modifications.

The paper is well written, with sufficient information of the methods and statistics. Overall, I think this is a noteworthy paper, well conducted and described, that is worth publishing. I don't see major shortcomings, so I can enthusiastically recommend it.

*Reviewer #2:*

This paper presents some very lovely high-resolution measurements of DNA unwrapping from various nucleosome configurations as well as Pol II transcribing through the same nucleosomes. The experimental method of pulling DNA off of the nucleosome by unwinding it from one end is very cool, though I have some concerns about the analysis methods to produce the so-called topographic map. The transcriptional map experiments seem very well done and seem like a very significant advancement. It seems like these are the most important measurements in the manuscript and they are mostly supported by the topography experiments (see my concerns about those below). It is not clear that the topography experiments themselves are necessary to interpret the transcription experiments. It is very exciting to see the distinct differences in the effects that H2A.Z vs. uH2B have on Pol II. The mechanical modeling seems sound. Overall, the paper is very strong and exciting and should be suitable to publication in *eLife* with the following analysis and interpretation concerns adequately addressed.

I have a serious concern regarding the analysis method used to construct the topographic maps (e.g., Figure 1). They point out that the dwell time for each force-vs.-extension contour reflects the strength of the local histone-DNA interaction. They also point out one complication: What really matters is the force where the transition occurs. The higher the transition force, yes, the longer the dwell time. But also the dwell time depends on where the force began, which depends on both the previous transition force and the amount of DNA released (gives the size of the resultant force drop). All the transitions are in a similar force range and the amount of released DNA vary by factors of 2-4. So it seems that the reported dwell times for a given rupture are just as dependent on how far the previous rupture dropped in force as they are on the magnitude of their own rupture force. I am concerned that this would create a strong convolution and severely scramble the conclusions drawn from the map. E.g., for Figure 1C, hWT unzipping, we see two distinct ruptures at very high force near 30 pN. The first rupture has a very long dwell time because the preceding rupture occurred at only 22 pN, while the second has a very short dwell despite needing also 30 pN to rupture because the preceding rupture also occurred at 30 pN. So while these two transitions which required similarly high force to rupture get scored by this method very differently, with the second appearing as a very weak interaction. It seems they are aware of the possibility of such an underestimation of the interaction. But the same effects also overestimate the first rupture. Why can they not measure the distribution of rupture forces (similar to protein folding force spectroscopy methods) directly? Otherwise, they should validate their analysis method on e.g., a long well modeled hairpin or provide some better justification that the conclusions from this method are not seriously flawed.

I think there is some unnecessary confusion in the descriptions of the initial force-extension method. From the initial presentation, it seems pulling on the NPS only is done very near equilibrium since it is shown that there is no hysteresis comparing pulling and relaxing. Later though, hopping is discussed and whether it is seen in the pulling curves and claim they pulled fast enough to not get hopping, i.e., out of equilibrium. But they say they do sometimes see hopping, for nucleosome pulling curves. This seems confusing and possibly contradictory. But either way, pulling curves should be shown plotted with sufficient bandwidth to say much about whether hopping does or doesn't occur. Hopping of bare hairpin DNA (no nucleosome) can be very fast, and the lack of hysteresis strongly suggests it is occurring faster than plotted. It would be helpful to the reader if they more clearly explained the pulling conditions and how the force ramp rate (pN/s) rather than trap displacement speed (nm/s) relates to the scale of unzipping and re-zipping times.

---

## [Author Response]

Reviewer #2:[…] I have a serious concern regarding the analysis method used to construct the topographic maps (e.g., Figure 1). They point out that the dwell time for each force-vs.-extension contour reflects the strength of the local histone-DNA interaction. They also point out one complication: What really matters is the force where the transition occurs. So the higher the transition force, yes, the longer the dwell time. But also the dwell time depends on where the force began, which depends on both the previous transition force and the amount of DNA released (gives the size of the resultant force drop). All the transitions are in a similar force range and the amount of released DNA vary by factors of 2-4. So it seems that the reported dwell times for a given rupture are just as dependent on how far the previous rupture dropped in force as they are on the magnitude of their own rupture force. I am concerned that this would create a strong convolution and severely scramble the conclusions drawn from the map. E.g., for Figure 1C, hWT unzipping, we see two distinct ruptures at very high force near 30 pN. The first rupture has a very long dwell time because the preceding rupture occurred at only 22 pN, while the second has a very short dwell despite needing also 30 pN to rupture because the preceding rupture also occurred at 30 pN. So while these two transitions which required similarly high force to rupture get scored by this method very differently, with the second appearing as a very weak interaction. It seems they are aware of the possibility of such an underestimation of the interaction. But the same effects also overestimate the first rupture. Why can they not measure the distribution of rupture forces (similar to protein folding force spectroscopy methods) directly? Otherwise, they should validate their analysis method on e.g., a long well modeled hairpin or provide some better justification that the conclusions from this method are not seriously flawed.

We thank the reviewer for this insightful comment. First, we would like to point out that, using residence time of the unzipping fork to map the position and strength of histone-DNA interactions in a nucleosome is a well-validated approach (Hall et al., 2009). Notably, in the Hall et al. paper, they acknowledged that “When two interactions occurred in close vicinity, upon the disruption of the first interaction the force was unable to relax back to the baseline before being ramped up again for the second interaction, subjecting this subsequent interaction to a higher initial force. Therefore, for each region of interactions, the dwell time histogram highlighted the edge of the region first encountered”.This effect is present regardless of pulling or loading rate.

Accordingly, in our analysis method, we did not measure the simple residence time, but rather the *force-weighted* residence times, which take into account the contribution of the forces at which the transitions occur. We have clarified this point in the manuscript, in Figure 1E, and in the Materials and methods section “Residence time analysis of unzipping traces”. Measuring the distribution of rupture forces requires some underlying assumption regarding the number of transitions and where they occur. As our data showed, these factors changed significantly when the histones were modified, especially with H2A.Z, and therefore comparing rupture forces would be problematic as the transitions compared would be different. Measuring force-weighted residence times allows for an unbiased characterization of the nucleosome unzipping process, and the positions of ruptures and the strength of the interactions at these positions arise directly from the data. For this reason, we preferred the force-weighted residence time approach. We believe that comparison to the transcription data is highly insightful, as the transcription traces display a similar pattern of pausing to the patterns observed with unzipping, as well as similar effects of histone modifications. These indicate that the direct mechanical barrier to unzipping (which is mainly due to position-specific histone-DNA contacts) plays a major role in forming the transcriptional barrier to the polymerase, and the similarity also strengthens our confidence in the residence time analysis.

The “force-weighted residence times” plotted in Figure 1E (force integrated over time spent dwelling at that specific base pair) correspond to the work required to unzip that base pair, given the constant trap separation rate in our experimental protocol. In the case of a system pulled slowly compared to its equilibration rates (as for bare DNA), this work should give an accurate measurement of the interaction energy landscape (details described in the Materials and methods; extraction of interaction landscape directly from bare DNA pulling curves is shown in Figure 3—figure supplement 1C, D, E). In the presence of nucleosomes, the trajectories obtained are, of necessity, out of equilibrium. In these situations, ruptures are stochastic events, and the dwell times are selected from a distribution which depends on the height of energetic barriers to unzipping, as well as the force at which the barrier is approached. By averaging over many trajectories, however, we believe the force-weighted dwell times gives an approximate, if not perfect, indicator of histone-DNA interaction strength. To test the validity of our force-weighted residence time analysis, we also collected constant force unzipping data (Figure 2). In these experiments, we raised the force to 28 pN and we kept it constant throughout the nucleosome unzipping process. In these experiments each barrier experienced the same force and the dwell times at each position are therefore comparable and a measure of the strength of the interaction between the DNA and the histone core, without the bias of earlier on later contacts. This alternative approach yielded a topographic map very similar to that obtained with the first method.

Furthermore, acknowledging the limitations of non-equilibrium measurements, in Figure 3 we used a distinct approach that allowed the system to explore its equilibrium energy landscape in the nucleosome entry regions. The measured equilibrium occupation probabilities are converted directly to interaction energies (Figure 3D and Figure 3—figure supplement 1). This technique yields a peak interaction energy that corresponds closely to one of the peaks observed in Figures 1E and 2B. Although the dwell time spectra of Figures 1E and 2B are fundamentally non-equilibrium, the differences in dwell times between different nucleosome types are still expected to reflect qualitative features of the specific DNA-nucleosome interactions.

To ensure that readers are aware of the limitations of the technique and data presented in Figures 1E and 2B, along with our discussion of those experimental results, we now write the following in the “Optical Tweezers assay for single-molecule” unzipping section:

“The pulling experiments used to obtain the average dwell times in Figures 1E and 2B are non-equilibrium in nature, reflecting an average over many out-of-equilibrium encounters with each nucleosome type, and are not direct indicators of equilibrium interaction strength alone. [...] Data from these pulling experiments is most useful when assessed in conjunction with data from other experimental techniques, such as the equilibrium hopping data in Figure 3, and the transcriptional residence times of Figures 6 and 7.”

I think there is some unnecessary confusion in the descriptions of the initial force-extension method. From the initial presentation, it seems pulling on the NPS only is done very near equilibrium since it is shown that there is no hysteresis comparing pulling and relaxing. Later though, hopping is discussed and whether it is seen in the pulling curves and claim they pulled fast enough to not get hopping, i.e., out of equilibrium. But they say they do sometimes see hopping, for nucleosome pulling curves. This seems confusing and possibly contradictory. But either way, pulling curves should be shown plotted with sufficient bandwidth to say much about whether hopping does or doesn't occur. Hopping of bare hairpin DNA (no nucleosome) can be very fast, and the lack of hysteresis strongly suggests it is occurring faster than plotted. It would be helpful to the reader if they more clearly explained the pulling conditions and how the force ramp rate (pN/s) rather than trap displacement speed (nm/s) relates to the scale of unzipping and rezipping times.

We apologize for the confusion. The reviewer raises an important point and we revised the main text. We deleted the sentence that caused the confusion: “Surprisingly, hopping, which is a hallmark of equilibration, was observed at strand separation rates expected to drive and keep the system out of equilibrium throughout unzipping.”, and replaced it with “The reversible nature of this transition contrasts with the irreversible transitions observed deeper in the nucleosome, and may indicate a rapid, small scale unzipping that is not large enough to disrupt the structure of the octamer. To better capture these hopping dynamics, […]”

In order to observe hopping, two requirements must be met: (1) there must be multiple well-defined states accessible at a given trap separation, where the rate of transitions between states are slow compared to the data collection frequency (800Hz in our experiments) and (2) the transition rates between these states must be sufficiently fast compared to the pulling rate to enable several back-and-forth transitions across a single barrier before subsequent unzipping occurs. We observe hopping only in the presence of the nucleosomes and not for the bare NPS, which indicates multiple states at a single trap separation for nucleosome-bound DNA. This is confirmed by our equilibrium measurements (constant trap separation) in Figure 3B. For bare DNA, most trap separations result in a single most likely position of the unzipping fork. In the presence of a bound nucleosome however, there can be multiple peaks of comparable height at certain trap separations, allowing for hopping of the unzipping fork between two or more states. The hopping behavior in the proximal nucleosome region is consistent with the nucleosome interaction energy profile extracted from equilibrium data (see Figure 3D). Given that the initial pulling experiments were clearly out-of-equilibrium (as evidenced by the hysteresis) the presence of hopping indicates that the barrier to the reverse transition is sufficiently low to allow such transitions to occur before the next transition was reached. This behavior is consistent with the nucleosome interaction energy profile extracted from equilibrium data (Figure 3D). The energetic barrier marked with a * leads to a relatively rapid reverse transition, while the subsequent barrier is very high. Such a system would thus hop repeatedly between two local minima until the force increases to high enough values to overcome the next large barrier.

In our pulling curves, the hopping behavior begins at roughly a 20pN pulling force and is no longer observed once the pulling force reaches 30pN. Given the pulling rate of 20 nm/sec and the trap stiffness of 0.3 pN/nm, the maximum force increase rate is 6 pN/sec. Of course, as the system relaxes under the force, this increase rate is a ceiling. From our force-extension data, we calculated the loading rate to be ~1.8-2.5 pN/sec. Using this approximate force increase rate, we would expect hopping to be observable if the transition time is below a couple of seconds. Our equilibrium data (Figure 3A) indicates that hopping at the proximal nucleosomal region occurs on a timescale of ~1 sec, consistent with the observable hopping in the constant extension pulling curves.

To relate the force ramp rate to the trap displacement speed, we now write:

“With nucleosome unzipping occurring at approximately 20-30 pN (Figure 1C and D), to observe hopping at our trap separation rate of 20 nm/s (with trap spring constant of approximately 0.3 pN/nm) requires the system to transition between states on timescales of approximately 2s or less.”

“Within an empirically determined trap distance range, we obtained equilibrium extension hopping traces (Figure 3A), which show transitions on timescales of approximately 1s, aligning with the necessary rate of transitions between states.”

It is worth noting that the ‘hopping’ we observed in the proximal dimer region of the nucleosome occurs much slower compared to that of bare hairpin DNA. We attribute these nucleosome-specific, very slow hopping events to local, reversible histone-DNA interactions. We also plotted one example force-extension curve of bare DNA versus nucleosome sample at maximum bandwidth (800 Hz) in Figure 3—figure supplement 1B to highlight this slow hopping event. To better explain the pulling condition, we also added the estimated loading rate of 1.8-2.5 pN/s in the Materials and methods section “Unzipping at a constant trap separation speed”.